# Learning General-Purpose Biomedical Volume Representations using Randomized Synthesis

**Neel Dey**[1]    **Benjamin Billot**[1]    **Hallee E. Wong**[1]    **Clinton J. Wang**[1]    **Mengwei Ren**[2]
**P. Ellen Grant**[3]    **Adrian V. Dalca**[1,3]    **Polina Golland**[1]
[1] MIT CSAIL    [2] New York University    [3] Harvard Medical School
`dey@csail.mit.edu`

## Abstract

Current volumetric biomedical foundation models struggle to generalize as public 3D datasets are small and do not cover the broad diversity of medical procedures, conditions, anatomical regions, and imaging protocols. We address this by creating a representation learning method that instead anticipates strong domain shifts at training time itself. We first propose a data engine that synthesizes highly variable training samples that would enable generalization to new biomedical contexts. To then train a single 3D network for any voxel-level task, we develop a contrastive learning method that pretrains the network to be stable against nuisance imaging variation simulated by the data engine, a key inductive bias for generalization. This network's features can be used as robust representations of input images for downstream tasks and its weights provide a strong, *dataset-agnostic* initialization for finetuning on new datasets. As a result, we set new standards across *both* multimodality registration and few-shot segmentation, a first for any 3D biomedical vision model, all without (pre-)training on any existing dataset of real images.

## 1 Introduction

Biomedical vision models trained on imaging studies with fixed protocols rarely generalize to new populations, medical procedures, and imaging devices. These domain shifts then necessitate practically infeasible reannotation and retraining cycles, especially for adaptation to new tasks. Further, *volumetric* annotated biomedical datasets are especially limited in sample size and focused on specific medical procedures, diseases, or scales of anatomy, leading current networks to overfit to a small subset of biomedical tasks. To overcome this data scarcity and heterogeneity, we present a representation learning framework driven by a synthetic data engine. Our approach yields a generalist 3D network that performs well on diverse voxel-level tasks across a range of unseen biomedical contexts in radiology.

Current biomedical foundation models are trained by aggregating publicly available datasets to cover multiple domains (Butoi et al., 2023; Chen et al., 2024; Liu et al., 2023a; MH Nguyen et al., 2024; Pachitariu & Stringer, 2022; Xie et al., 2022). However, persistent gaps hinder their widespread adoption. For example, some methods operate only on specific modalities and regions that can be tractably scaled up in sample size, such as abdominal CT (Hamamci et al., 2024; Li et al., 2024a) , and thus cannot learn general representations for most other domains, such as *in utero* fetal MRI. Others treat 2D slices of volumetric images as independent data points (Butoi et al., 2023; Ma & Wang, 2023), often constructing training sets with high inter-sample correlation and yielding models that fail to produce consistent 3D results. Further, existing foundation models almost exclusively focus on segmentation and classification and neglect other key vision tasks such as registration. To our knowledge, no biomedical vision foundation model has been demonstrated for multiple disparate 3D tasks yet.

**Contributions.** This paper makes advances on two fronts. To gain robustness to large domain shifts in downstream deployment, we first propose a biomedically informed data engine whose samples encompass a wide range of appearances and semantics. This engine uses randomly sampled spatial configurations of biomedical shape templates to synthesize images with arbitrary resolutions, appearances, imaging physics, and crucially, minimal influence from any existing biomedical dataset. Unlike training on samples from GANs or diffusion models, which are limited to reproducing only their original training distribution, our engine synthesizes highly diverse samples useful for *arbitrarily new* biomed-

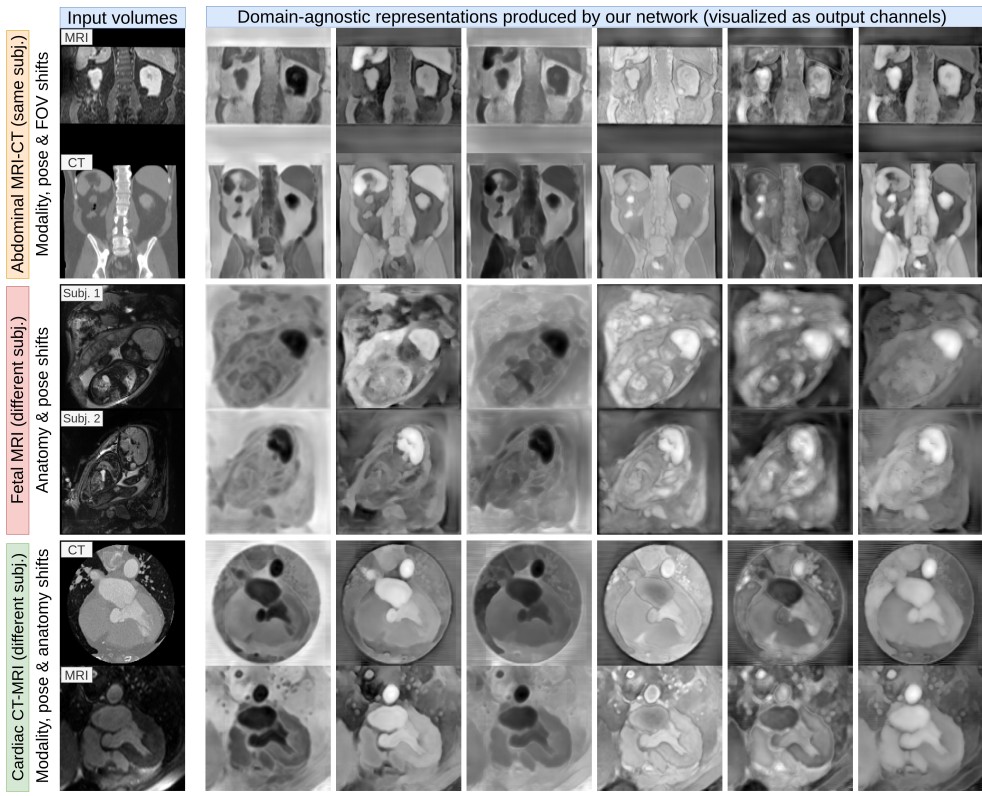

Figure 1: **Representations produced by our framework**, trained only on synthetic data, are approximately stable across imaging modalities, field-of-views, and poses on real unseen volumes from various datasets. For each anatomical region (**rows**), we show two example volumes with substantial variation (**col. 1**) and six arbitrarily selected network output channels (**cols. 2–7**) that illustrate this stability. These features and network weights can be used for several voxel-level tasks.

ical contexts that we do not have training data for. We then develop a contrastive learning framework that uses paired samples from the data engine to pretrain a network for general voxel-level tasks in radiology using an inductive bias of approximate stability to nuisance imaging variation that does not change image semantics, a key property for generalization across datasets (Gruver et al., 2022).

Our experiments demonstrate that the resulting features and weights enable broad generalization on the key biomedical tasks of 3D registration and segmentation across several diverse datasets. We achieve state-of-the-art unsupervised multimodality image registration by simply using the network's approximately appearance invariant and pose equivariant representations (Fig. 1) to drive existing registration solvers. The proposed network can also be used as an off-the-shelf *dataset-agnostic* initialization for finetuning on any voxel-level task. Specifically, we demonstrate strong few-shot segmentation performance across several highly variable downstream datasets, thereby removing the need for the cumbersome dataset-specific self-supervised pretraining commonly used today. Code, tutorials, and model weights are available at `https://www.neeldey.com/anatomix/`.

## 2 RELATED WORK

**Generative image models.** Learning-based generative models (Brock et al., 2018; Goodfellow et al., 2014; Karras et al., 2020; Luo, 2022; Rombach et al., 2022; Song et al., 2020) trained on internet-scale natural vision sets (Schuhmann et al., 2022) can now synthesize photorealistic samples for pretraining general networks (Donahue & Simonyan, 2019; Fan et al., 2023; Li et al., 2023; Tian et al., 2024b). However, such generative models trained instead on the few thousand publicly available anatomy- and modality-specific annotated volumes in biomedical datasets (Baid et al., 2021; LaMontagne et al., 2019; Qu et al., 2024; Wasserthal et al., 2023) generally do not learn representations that can

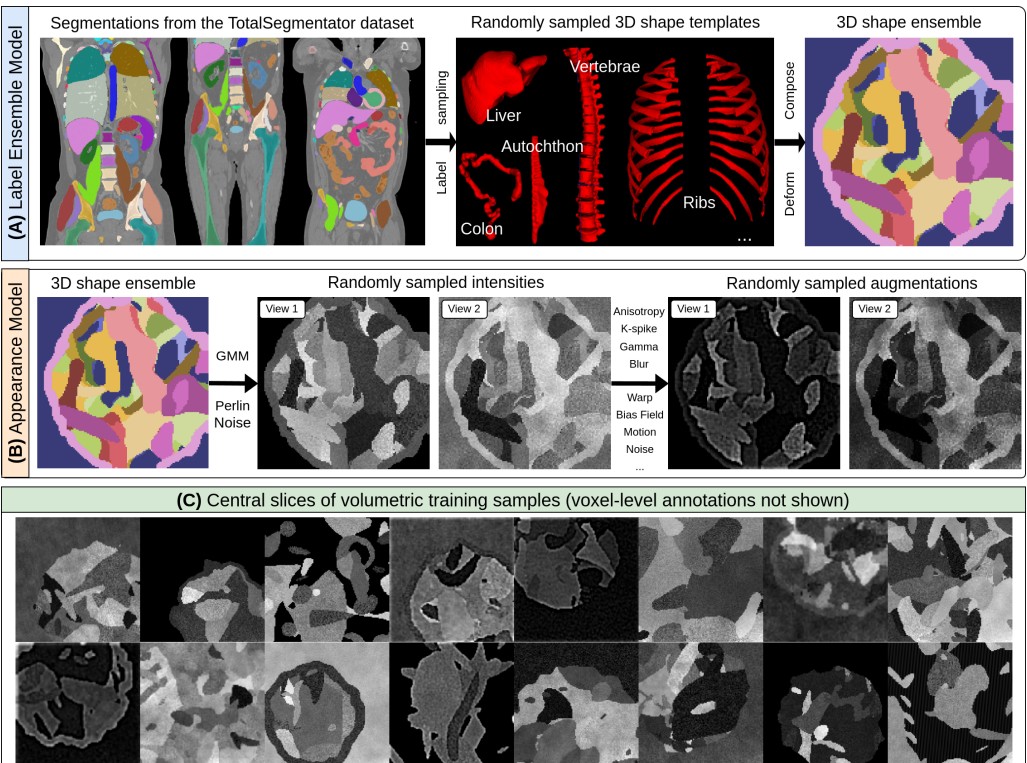

Figure 2: **Data engine. A.** We randomly sample binary labels as templates from a large database of anatomical segmentations to create 3D label ensembles of randomly deformed templates. **B.** Given a synthesized label ensemble and an appearance model, we synthesize two volumes to pretrain a network with a dense multi-view contrastive objective. **C.** Example synthetic training volumes produced by our data engine. These samples are not intended to be necessarily realistic, but rather to serve as diverse and useful training data for learning general tasks in arbitrary radiological domains.

generalize to new biomedical domains. In contrast, the label synthesis component of our data engine draws loose inspiration from the Dead Leaves model (Baradad Jurjo et al., 2021; Lee et al., 2001). This hand-crafted generative model considers images to be compositions of randomly deformed shape templates (such as cubes, ellipsoids, etc.) with static intensities to capture the statistics of natural images. We compose biomedical shape templates similarly but develop several further extensions and propose a distinct appearance model, as explained below.

**Domain randomization.** To learn robustness to domain shifts at deployment, domain randomized generative models (Tobin et al., 2017) trade realism for diversity when generating training data for downstream models. For example, domain randomized methods for brain segmentation (Billot et al., 2023a;b; Gopinath et al., 2023; Hoopes et al., 2022b) and registration (Hoffmann et al., 2021; Iglesias, 2023) train on synthetic brains simulated from label maps and require large collections of expert brain annotations. Recent work in dataset-agnostic instance segmentation (Chollet et al., 2024; Dey et al., 2024) generalizes beyond brains by simulating both annotations and images using a pre-specified shape prior. We build on these concepts to develop a *task-agnostic* data engine that simulates images with highly variable appearances and physics using compositions of biomedical shape templates.

**Invariant imaging features.** Given the heterogeneous nature of biomedical imaging protocols, several existing strategies aim to extract features robust to nuisance variation. When registering images across modalities, aligning modality-invariant hand-crafted local descriptors (Heinrich et al., 2012; 2013) and/or edges (Haber & Modersitzki, 2006) is common. With deep learning-based multimodality registration, this inter-modality invariance can be learned (Dey et al., 2022; Mok et al., 2024; Pielawski et al., 2020), leading to improved performance at the cost of dataset-specific training. Beyond registration, brain-specific invariant features have been learned by exploiting large repositories of annotated brains (Chua & Dalca, 2023; Liu et al., 2023b; 2024). Our work obviates

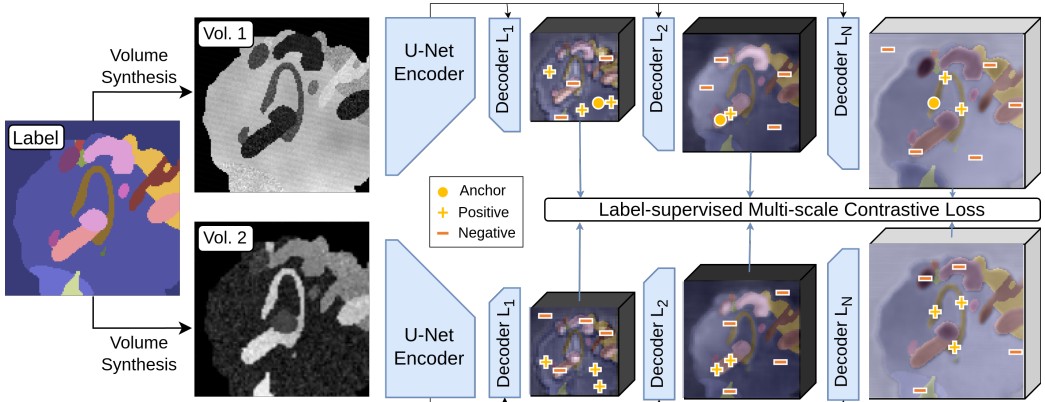

Figure 3: **Representation learning.** Given a 3D label map and two corresponding synthetic volumes sampled from our data engine, we process them using a single UNet with shared weights. The UNet is pretrained contrastively at each decoder layer: for a randomly sampled anchor, features sampled from the same label in both volumes serve as positives, while features from other labels act as negatives.

the need for dataset-specific training, anatomical region-specific modeling, and large-scale annotation collection by extracting modality- and appearance-invariant features in an amortized manner.

**Volumetric pretraining.** Many methods pretrain on unannotated images from a dataset before supervised finetuning on a small labeled subset. These methods often employ self-supervised reconstructive (Chen et al., 2019a; Tang et al., 2022; Valanarasu et al., 2024; Zhou et al., 2023; 2021) and/or discriminative (Chaitanya et al., 2020; 2023; Ren et al., 2022; You et al., 2024) losses. However, these pretraining strategies exploit heuristics about their target datasets that often do not broadly generalize (Dong et al., 2021; Ren et al., 2022). Our approach instead yields a network that generalizes to arbitrary radiological datasets and does not require bespoke pretraining frameworks for each project. Lastly, recent biomedical foundation models trained on pooled datasets of 2D slices (Butoi et al., 2023; Ma & Wang, 2023; MH Nguyen et al., 2024; Wong et al., 2023) generally require interaction (via bounding boxes, scribbles, etc.), struggle with 3D consistency, and are restricted to segmentation. We instead directly train for general tasks using synthetic 3D volumes and do not work on interactive tasks.

## 3 METHODS

This section first details the proposed data engine (Fig. 2), then describes the representation learning strategy (Fig. 3), and concludes with applications towards 3D registration and segmentation.

**Data engine: label ensemble model (Fig. 2A).** We create synthetic 3D label ensemble volumes by sampling from a repository of biomedical shape templates. As templates, we use the freely available $\sim$45,000 binary volumes from the TotalSegmentator dataset of 104 annotated organs in 1,204 CT volumes (Wasserthal et al., 2023). For each label ensemble volume, we iteratively populate a 3D volume with a random number of randomly sampled templates that are each then deformed and assigned a label corresponding to the sampling iteration. As radiological volumes are often surrounded by empty space (e.g., air), we simulate foreground and background separation by applying a binary foreground mask to two-thirds of the synthesized label ensembles by multiplying them with a randomly deformed binary sphere with a random radius and center. Further, as many radiological applications and domains have layer-like structures at the foreground-background interface (e.g., fat and skin), we randomly encase half of the foreground-masked volumes within envelope labels of random widths.

**Data engine: appearance model (Fig. 2B).** Given a 3D label ensemble $L$ with $K$ labels, we sample the intensities of two volumes $V_1$ and $V_2$ from two independent $K$-component Gaussian mixture models (GMMs) each with parameters $\{\mu_{k1}, \sigma_{k1}^2\}_{k=1}^K$ and $\{\mu_{i2}, \sigma_{i2}^2\}_{k=1}^K$, respectively, all of which are randomly drawn from uniform distributions. For each spatial index in $L$ with label $k$, we sample the initial intensities in $V_1$ and $V_2$ from $\mathcal{N}(\mu_{k1}, \sigma_{k1}^2)$ and $\mathcal{N}(\mu_{k2}, \sigma_{k2}^2)$, respectively. We then pointwise multiply them with Perlin-like noise (Perlin, 1985) to simulate spatial texture and augment

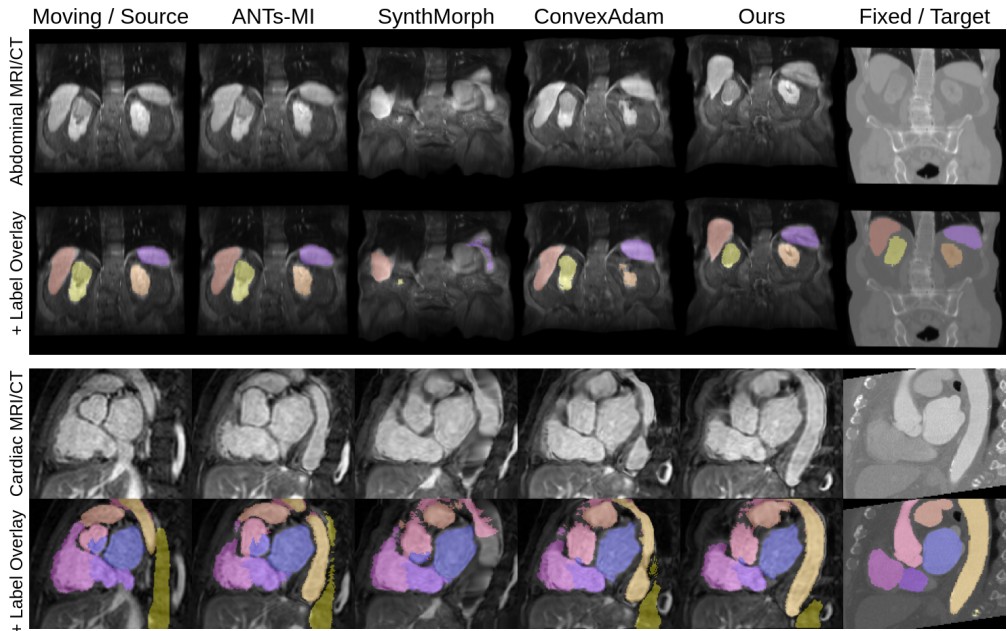

Figure 4: **Multi-modality 3D registration.** Using our representations with the ConvexAdam registration solver ("**Ours**") yields accurate alignment of challenging pairs of intra-subject abdominal MRI-CT (top) and inter-subject cardiac MRI-CT (bottom) with large deformations across modalities.

using transformations relevant to biomedical volumes such as random bias fields, Fourier-spikes, Gamma shifts, blurring, Gibbs ringing, resolution degradations, noise, motion, flips, and affine warps. All intensity augmentations are sampled independently but the geometric augmentations are shared.

In summary, we synthesize a 3D label ensemble $L$ and draw volumes $V_1$ and $V_2$ from it that differ in appearance but share 3D semantic layouts. This is repeated with randomized hyperparameters for each sample to generate a synthetic dataset. Low-level modeling details are provided in App. B.1.

**Contrastive pretraining (Fig. 3).** As our data engine provides exact label supervision, we develop a representation learning loss that is a spatial extension of multi-positive supervised contrastive learning (Khosla et al., 2020). To pretrain network $F : \mathbb{R}^{H \times W \times D} \to \mathbb{R}^{H \times W \times D \times C}$ where $H, W,$ and $D$ are spatial dimensions and $C$ is the number of output features, we use an inductive bias of voxel-level features within a 3D shape having similar spatial representations, regardless of differences in appearance. We assume that an anchor spatial index $i \in I$ (where $I = \{1, \ldots, 2HWD\}$, i.e., voxels pooled from $V_1$ and $V_2$) with features $f_i \in \mathbb{R}^C$ in label $k$ should have similar representations to all other indices in label $k$ in both $F(V_1)$ and $F(V_2)$ and dissimilar representations to indices from other labels. As in Chen et al. (2020a), we use a non-linear projection $Z : \mathbb{R}^{H \times W \times D \times C} \to \mathbb{R}^{H \times W \times D \times C_Z}$ on $F$'s outputs followed by an $L_2$-normalization when computing the contrastive loss,

$$\mathcal{L} = \sum_{i \in I} \frac{-1}{|P(i)|} \sum_{p \in P(i)} \log \frac{\exp(z_i \cdot z_p / \tau)}{\sum_{q \in Q(i)} \exp(z_i \cdot z_q / \tau)}, \tag{1}$$

where $z \in \mathbb{R}^{C_Z}$, $\tau$ is a hyperparameter, $Q(i) = I \backslash \{i\}$ (i.e., all non-anchor spatial indices), and $P(i)$ is the set of all positives for anchor $i$, s.t. $P(i) = \{p \in Q(i) : k_p = k_i\}$, where $k_x$ is the label of spatial index $x$. Lastly, we use this loss on multiple decoder layers of $F$ during training to leverage multiscale (self-)supervision as in Dey et al. (2022); Park et al. (2020); Ren et al. (2022).

**Pretraining implementation details**. We implement $F$ as a four-level 3D convolutional UNet (Ronneberger et al., 2015) following the architecture from (Ren et al., 2022) and construct $Z$ as a 3-layer 128-node-wide MLP. While $F$ can be any volume-to-volume network, we use a U-Net as it is the standard architecture for biomedical imaging tasks and performs well across datasets (Isensee et al., 2024; Stringer & Pachitariu, 2024). $F$ and $Z$ are pretrained jointly for 600,000 iterations with a batch size of one $128^3$ label map, each generating two $128^3$ volumes. We compute the contrastive

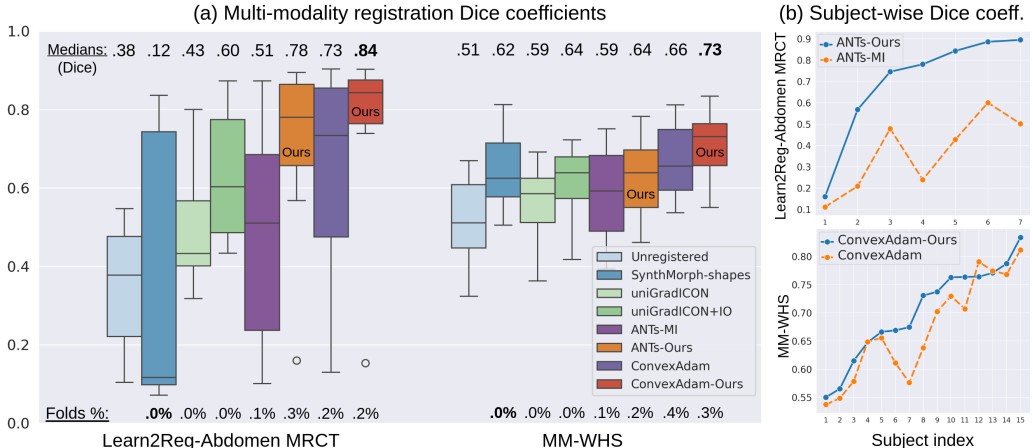

Figure 5: **Multi-modality 3D registration results. (a)** Dice boxplots for each method for L2RAb (**left group**) and MM-WHS (**right group**), with corresponding medians reported on **top** of each box and the mean percentages of voxels with folds produced by each method reported at the **bottom**; **(b)** Using our features leads to consistent registration improvements at the subject-level.

loss (with temperature $\tau = 0.33$) on 512 randomly sampled indices at each iteration for each decoder layer due to memory limitations. Lastly, $Z$ is only used during pretraining and is discarded for all downstream tasks. All other pretraining details are described in App. B.3.

**Downstream tasks: multi-modality registration.** Gradient-based deformable registration objectives typically take the form of $\mathcal{L}_{reg} = d(V_{\text{fixed}}, V_{\text{moving}} \circ \varphi) + \lambda \text{Reg}(\varphi)$, where image $V_{\text{moving}}$ is to be aligned to $V_{\text{fixed}}$ by deformation $\varphi$, subject to regularization $\text{Reg}(\cdot)$, and $d(\cdot)$ is an image dissimilarity score. To align images across imaging modalities, we simply replace the input volumes with our pretrained network's features as in $\mathcal{L}_{reg} = d(F(V_{\text{fixed}}), F(V_{\text{moving}}) \circ \varphi) + \lambda \text{Reg}(\varphi)$. This approach is compatible with any existing high-performance registration solver, such as the ConvexAdam (Siebert et al., 2021; 2024) and ANTs (Tustison et al., 2020) frameworks used in our experiments below.

**Downstream tasks: few-shot segmentation.** For $N$-label few-shot semantic segmentation, we finetune the pretrained network $F$ on a small number of annotated volumes with an additional convolutional layer with softmax activation and $N$ output channels. We optimize the network using an equally weighted sum of the soft Dice and cross-entropy losses (Isensee et al., 2021; Taghanaki et al., 2019). To extract strong performance for all baselines in the challenging setting of finetuning on only one or few annotated volumes, we use extensively-tuned augmentation pipelines and finetune *all layers* of each network for a high number of iterations (37,500) with cosine learning rate decay.

## 4 EXPERIMENTS

The pretrained network's output features for inter and intra-subject volume pairs with semantically similar content subject to strong domain shifts are visualized in Fig. 1. Below, we present experiments that investigate the utility of our learned representations for multi-modality registration, the network weights as a pretrained initialization for few-shot segmentation, and analyze our modeling decisions.

### 4.1 UNSUPERVISED MULTI-MODALITY DEFORMABLE 3D REGISTRATION

**Data and setup.** We use the Learn2Reg AbdomenMRCT (Hering et al., 2021) (L2RAb) and MM-WHS (Gao et al., 2023; Zhuang, 2018; Zhuang et al., 2019) datasets to benchmark MRI to CT volume registration. L2RAb is an abdominal registration benchmarking dataset, whose publicly available portion provides eight affine-aligned intra-subject MRI and CT pairs of size $192 \times 160 \times 192$ at $2 \times 2 \times 2\text{mm}^3$ resolution, with labels for four organs. MM-WHS, originally a heart segmentation dataset, contains 20 annotated MRIs and CTs (from distinct subjects) and we affine-align all volumes to a common space of size $160 \times 160 \times 128$ at $1.142 \times 1.142 \times 1.283\text{mm}^3$ resolution (see App. B.4.4 for further details). The registration experiments are unsupervised and

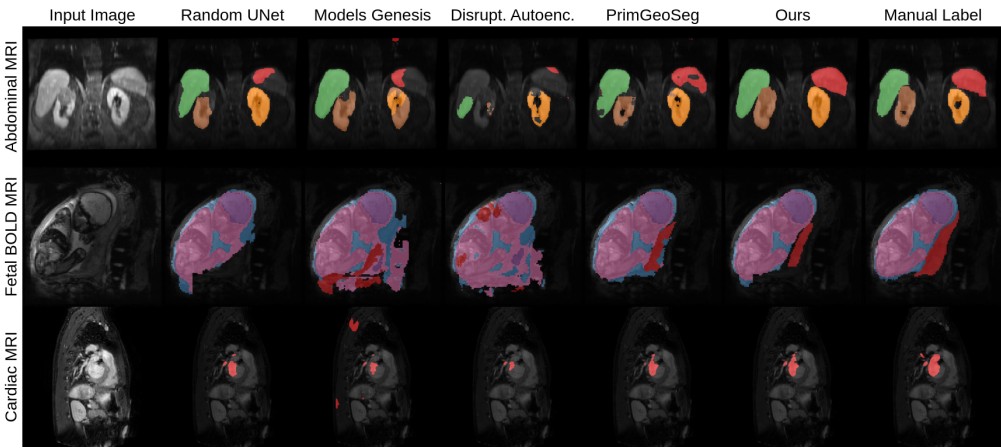

Figure 6: **Few-shot 3D segmentation qualitative results**. All methods (**columns 2–7**) were fine-tuned on 3, 3, and 1 multi-label annotated volume(s) for each respective dataset (**rows 1–3**).

we split L2RAb and MM-WHS into 1/7 and 5/15 validation/testing pairs, respectively, where the validation pair(s) are only used for tuning registration hyperparameters.

**Baselines and evaluation.** We compare unsupervised and training-free multimodality registration frameworks. Our iterative baselines include the widely-used `ANTs` library (Avants et al., 2008; Tustison et al., 2020) with mutual information loss (Mattes et al., 2001; Wells III et al., 1996) (`ANTs-MI`) and the state-of-the-art (Hering et al., 2021) multimodality method, `ConvexAdam` (Siebert et al., 2021). We further use two dataset-agnostic registration networks, `SynthMorph-shapes` (Hoffmann et al., 2021) and `uniGradICON` (Tian et al., 2024a), with the latter also using optional instance optimization (`uniGradICON+IO`). For evaluation, we report post-registration volume overlap (Dice) of anatomical structures. We also assess deformation inverse consistency using the percentage of folding voxels, where $\det(J_\varphi) < 0$, for Jacobian $J_\varphi$ of the estimated warp $\varphi$. Folding percentages below 0.5% of all voxels are generally considered negligible and methods producing higher Dice values while staying under this threshold are preferred (Qiu et al., 2021).

**Adaptation to use network features.** We modify `ANTs` and `ConvexAdam` to use our pretrained network's 16 extracted features. We use `ANTs`' multichannel mode with the MSE loss and tune its hyperparameters heuristically on validation pairs. For `ConvexAdam`, we concatenate our features with its default handcrafted features and perform a grid search for both the original implementation and our variant over four hyperparameters. Lastly, the deep learning baselines assume single-channel input volumes and thus cannot directly use our multichannel network features.

**Results.** Figs. 4 and 5 present results on held-out testing pairs. `ANTS-Ours` strongly improves upon the typically-used `ANTS-MI`, with 26 and 5 points of median Dice improvement on L2RAb and MM-WHS, respectively, while maintaining nearly identical low folding characteristics. Further, driven by our network features, `ConvexAdam-Ours` outperforms all methods in terms of volume overlap and improves on its base `ConvexAdam` method by 11 and 6 Dice points, with the same folding ratios. In contrast, the `SynthMorph-shapes` and `uniGradICON+IO` methods perform well on MM-WHS (where all hearts are roughly centered) but cannot handle the larger deformations in L2RAb. They do, however, yield nearly diffeomorphic transformations, producing almost zero folds. Lastly, without iterative optimization, `uniGradICON` demonstrates limited generalization across large intensity-based domain gaps. We note that all methods produce folding percentages that are under the threshold of 0.5% folding voxels. Additional grid search results on the validation sets are in App. A.2.

### 4.2 FEW-SHOT 3D MULTI-LABEL SEMANTIC SEGMENTATION

**Data and setup.** We evaluate few-shot segmentation performance on a diverse collection of datasets: cardiac bSSFP MRI from MSD-Heart (Antonelli et al., 2022), abdominal CT from AMOS (Ji et al., 2022), prostate T2w MRI from PROMISE12 (Litjens et al., 2014), abdominal SPIR MRI (Akin et al., 2016; Clark et al., 2013; Kavur et al., 2019; Linehan et al., 2016) from Learn2Reg-Abdomen (Hering

Table 1: **Few-shot 3D segmentation Dice** means and their bootstrapped std. deviations. Bolding and underlining represent the best and second-best Dice, respectively.

| | Params. | MSD-Heart | PROMISE12 | L2RAb-MRI | FeTA | AMOS-CT | WUFetal |
|---|---|---|---|---|---|---|---|
| Fine-tuning vols. | | 1 | 2 | 3 | 3 | 3 | 3 |
| Number of classes | | 1 | 1 | 4 | 7 | 15 | 4 |
| Rand. Init. UNet | 5.9M | .85(.01) | .80(.02) | .85(.06) | .78(.03) | .56(.01) | .73(.02) |
| Transfer Learning | 5.9M | .87(.02) | .82(.01) | .82(.06) | .78(.03) | .52(.01) | .74(.02) |
| Models Genesis | 19.1M | .84(.04) | .73(.03) | .81(.06) | .79(.03) | .55(.01) | .66(.03) |
| MedicalNet | 17.3M | .86(.02) | .53(.04) | .73(.07) | .74(.04) | .44(.02) | .50(.02) |
| PrimGeoSeg | 67.2M | .87(.01) | .79(.02) | .84(.05) | .79(.03) | **.63(.01)** | .76(.02) |
| SMIT | 67.2M | .88(.02) | .72(.03) | .84(.06) | .76(.04) | .58(.01) | .73(.02) |
| Disruptive AE | 67.2M | .82(.02) | .64(.03) | .77(.07) | .74(.04) | .50(.02) | .70(.02) |
| Ours | 5.9M | **.89(.01)** | **.85(.01)** | **.86(.06)** | **.80(.03)** | .61(.01) | **.76(.02)** |
| Full supervision | 5.9M | .91(.01) | .90(.00) | .89(.05) | .83(.01) | .85(.00) | .88(.01) |

Table 2: **Multitask capabilities of current 3D biomedical foundation models.** Using foundation models as feature extractors for multimodality registration with the `ANTs` solver, only `Ours` outperforms the solver defaults (`MutualInfo`), indicating that other methods are limited to segmentation.

| Dataset | MutualInfo | PrimGeoSeg | ModelsGen. | SMIT | DAE | **Ours** |
|---|---|---|---|---|---|---|
| L2RAbdomenMRCT | .48(.10) | .46(.09) | .38(.07) | .48(.08) | .50(.07) | **.70(.09)** |
| MM-WHS | .58(.03) | .51(.02) | .51(.03) | .53(.03) | .53(.03) | **.63(.02)** |

et al., 2021), fetal brain HASTE MRI from FeTA (Payette et al., 2024), and an in-house dataset of whole uterus fetal BOLD MRI (WUFetal). WUFetal provides labels for the placenta, amniotic fluid, and the fetal brain and body. It is included as an out-of-distribution test dataset for our baselines below that have not been pretrained on fetal images. Lastly, we operate in the few-shot regime, where only 1–3 annotated volumes per dataset are used for finetuning. The dataset splits are in App. B.5.

**Baselines.** We use 3D foundation models specifically pretrained for multi-label segmentation on multiple datasets. These include the masked autoencoding-based `Models Genesis` (Zhou et al., 2021), `SMIT` (Jiang et al., 2022), and `Disruptive AE` (Valanarasu et al., 2024). Other transfer-learning baselines include `MedicalNet` (Chen et al., 2019b) and `PrimGeoSeg`, the latter being pretrained to segment synthetic binary volumes with simplistic shapes. To explicitly test transfer learning with a matched architecture (`TransferLearning`), we further train a fully supervised UNet with the same architecture as `Ours` on a large-scale neuroimage segmentation dataset (Hoopes et al., 2022a; LaMontagne et al., 2019). We also test a randomly initialized UNet (with matched architecture to `Ours`) trained on few (`RandInitUNet`) or all (`FullSupervision`) volumes in the training split. Current 2D interactive binary segmentation foundation models (Butoi et al., 2023; Ma & Wang, 2023) were excluded from the experiments as they require user prompts and do not apply to 3D multi-label data. All methods were finetuned with extensive data augmentation, as described in App. B.5.

**Results.** Table 1 and Fig. 6 present few-shot segmentation results. Our framework produces pretrained weights that consistently improve upon random initialization. Crucially, this improvement is achieved using only our dataset-agnostic pretrained weights and without any pretraining on unlabeled real volumes or data from similar domains, as commonly done in previous work. Compared to larger foundation models specifically trained for segmentation, our pretrained general-purpose network achieves the first or second rank consistently and has the best average rank. Importantly, the second-best model changes dataset-to-dataset, indicating that the baseline methods do not generalize consistently to new biomedical contexts. We lastly emphasize that we achieve these gains with fewer parameters, without access to any real images, and while being applicable to other tasks as well, as described below.

## 4.3 ABLATIONS AND OTHER ANALYSES

**Multitask capabilities.** Current 3D biomedical vision foundation models are evaluated primarily for semantic segmentation, raising the question of how well their features generalize to other tasks. To

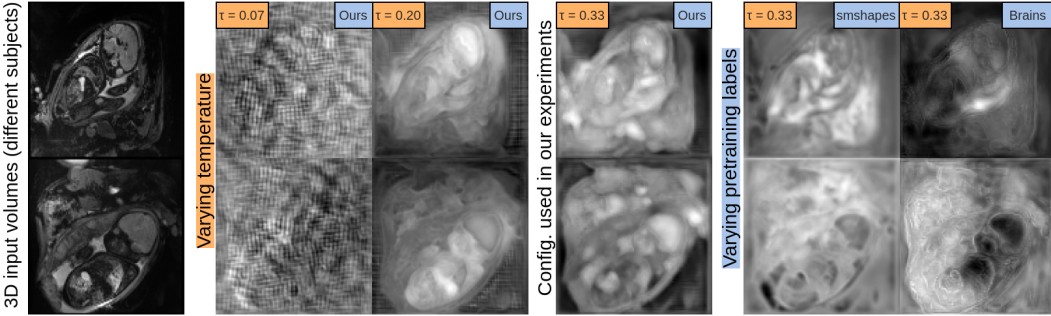

Figure 7: **Features for varying pretraining configurations.** When our framework is trained with different $\tau$ values (cols. 2–3) or on synthetic data generated from other label sources (cols. 5–6), the network features for real biomedical volumes (col. 1) are degenerate and/or sensitive to nuisance imaging variation. We visualize an arbitrary channel for each model, with more in App. Fig. 11.

test this, we utilize the few-shot segmentation baselines that provide both pretrained encoders and decoders as general-purpose feature extractors and use their features to drive a generic registration solver (`ANTs`), with implementation details provided in App. B.4.3. As reported in Table 2, all current 3D biomedical foundation models are unable to extract features that improve upon the baseline setting used by the default ANTs solver (`MutualInfo`). In contrast, our approach (`Ours`) outperforms it by wide margins and is thus the only method to yield multitask general-purpose features for the highly disparate tasks of multi-modality registration (Fig. 5) and few-shot segmentation (Table 1).

**Label generation.** To evaluate our label ensemble model, we replace it with other training labels while keeping the appearance model fixed. We test pretraining using: (a) synthetically generated labels that have no biomedical priors (Hoffmann et al., 2021) (`smshapes`), (b) 1,573 label maps of real brain MRI annotated using the FreeSurfer protocol (Fischl, 2012) (`Brains`) that represent real anatomical structures with dense per-voxel annotations, and (c) a combination of `Brains` and our model to mix real and synthetic sources of label maps. Details regarding these models are provided in App. B.6.

Table 3: **Effect of pretraining configurations on downstream tasks** via Dice means and their bootstrapped std. deviations. **Row 1** corresponds to the configuration used in our previous experiments. Registration experiments are all performed using the `ConvexAdam` (Siebert et al., 2021) solver.

| Pretraining config. | | | Registration (L2RAb) | | Few-shot segmentation Dice (↑) | | |
|---|---|---|---|---|---|---|---|
| Pretraining loss | $\tau$ | Labels | Dice(↑) | Folds%(↓) | WUFetal | MSD-Heart | AMOS-CT |
| Ours | 0.33 | Ours | **.74**(.10) | 0.22% | .76(.02) | .89(.01) | .61(.01) |
| *Ablating pretraining labels* | | | | | | | |
| Ours | 0.33 | smshapes | .68(.10) | 0.20% | .73(.02) | .88(.01) | .60(.01) |
| Ours | 0.33 | Brains | .57(.10) | 0.16% | .74(.02) | .88(.02) | .60(.01) |
| Ours | 0.33 | Ours+Brains | .71(.08) | 0.29% | .76(.02) | .89(.01) | .61(.01) |
| *Temperature variation* | | | | | | | |
| Ours | 0.07 | Ours | .58(.10) | 0.17% | .72(.02) | .90(.01) | .60(.01) |
| Ours | 0.20 | Ours | .64(.09) | 0.19% | **.78(.02)** | **.91(.01)** | **.62(.01)** |
| *Ablating pretraining loss* | | | | | | | |
| Denoising | - | Ours | .51(.10) | 0.19% | .58(.02) | .83(.03) | .46(.01) |
| Remove labels | - | Ours | .50(.11) | 0.24% | .66(.02) | .86(.02) | .58(.01) |
| *Ablating pretraining augmentations* | | | | | | | |
| Ours (Row 1) w/o FG mask | | | .63(.09) | 0.27% | .73(.02) | .86(.03) | .60(.01) |
| Ours w/o FG mask, w/o offline augm. | | | .63(.09) | 0.22% | .74(.02) | .89(.01) | .56(.01) |
| Ours w/o FG mask, w/o all augm. | | | .59(.09) | 0.63% | .70(.02) | .84(.02) | .51(.01) |

Table 3 rows 1–4 show that both registration and segmentation performance decline with other choices of pretraining labels. Further, combining our labels with `Brains` does not affect segmentation but worsens abdominal registration. Lastly, pretraining on these alternative label models leads to unstable network features on real data (Fig. 7, right), likely explaining the performance degradation.

**Temperature ($\tau$).** The temperature hyperparameter $\tau$ in the contrastive loss (eq. 1) controls the penalty weight on negative pairs (Wang & Liu, 2021). In natural vision, smaller $\tau$ values (such as $\tau = 0.07$ (Chen et al., 2020b)) are used to upweigh difficult negative pairs in the loss. However, as we train our network to learn similar representations within each label across highly disparate appearances, the negative pairs are all of higher difficulty and require a relaxed $\tau$ of 0.33 for stable training. We find that using lower $\tau$ leads to degenerate aliased features (Fig. 7) and worse registration results (Table 3). Interestingly, segmentation benefits slightly from an intermediate setting of $\tau = 0.20$, indicating a tradeoff between optimal representations for downstream registration versus segmentation.

**Pretraining objectives.** We now retain our data engine, but pretrain using other frameworks. We compare against matched architectures that are (a) pretrained to denoise the augmentations used in our data engine (`Denoising`) as in Iglesias et al. (2023) and (b) pretrained using self-supervision on intensities alone without using label information (`RemoveLabels`) as in Chua & Dalca (2023); Ren et al. (2022). In Table 3, rows 1, 7, and 8, we find that our label-supervised multi-positive contrastive strategy yields the highest results for both registration and segmentation.

**Data engine augmentations.** We now ablate the augmentations used in the proposed data engine used during pretraining by cumulatively removing the foreground masking, the augmentations used during offline image synthesis (App. Fig. 12), and all augmentations, such that the training images are simply the Perlin-corrupted Gaussian mixture model outputs. In Table 3 rows 9–11, we observe a substantial drop in both registration and segmentation performance without the proposed augmentations.

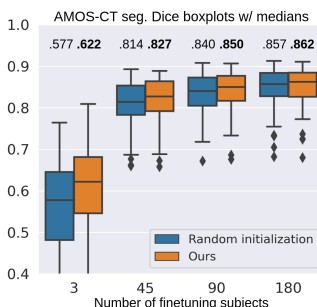

Figure 8: Fine-tuning performance as a function of annotation budget.

**Finetuning with more annotations.** Finally, while our segmentation experiments focus on the few-shot setting, our learned initialization also benefits scenarios with more supervision. Fig. 8 quantifies how finetuning the proposed network with more annotated volumes leads to improved segmentation on AMOS-CT (Ji et al., 2022) relative to random initialization across settings, albeit with smaller improvements given more annotations.

## 5 DISCUSSION

**Limitations and future work.** Our approach does have limitations. We pretrained our network to be stable against intensity variations (among other variables) and demonstrated its utility for registration and segmentation. However, a small set of biomedical tasks *rely* on relative intensities (Nakamura et al., 2017; Thomalla et al., 2011) and texture (Yu et al., 2017), making intensity invariance a suboptimal inductive bias for them. Further, while we operate on general volumetric tasks, certain inverse problems like MRI reconstruction take sensor-domain non-Cartesian (Schlemper et al., 2019) measurements as multichannel inputs, requiring domain-specific architectural changes (Singh et al., 2022). Lastly, our segmentation experiments finetune our pretrained network on specific datasets, potentially introducing complexity for some clinical users. However, future extensions could directly use our proposed data engine to train promptable 3D segmentation models that require no such finetuning.

**Conclusions.** When combined with the right inductive biases, synthetic data models informed by biomedical templates enable the training of powerful general-purpose networks. This is important for 3D radiology, where existing annotated datasets are limited in sample size, often acquiring only dozens to at most a few thousand volumes. This leads to inflexible models that deteriorate under domain shifts. Trained only on synthetic volumes with our proposed framework, the resulting network provides substantial benefits on a variety of radiological tasks. For example, its representations yield substantial improvements over the state-of-the-art in training-free multimodality deformable registration, a key area in biomedical vision. Further, the network can also serve as a downstream dataset-agnostic initialization for few-shot segmentation tasks and lead to improvements across multiple datasets.

## ACKNOWLEDGEMENTS

We gratefully acknowledge support from NIH NICHD R01HD100009, NIH NICHD 1R01HD114338, NIH NIBIB 1R01EB032708, NIH NIBIB R01EB033773, MIT-IBM Watson AI Lab, MIT Jameel Clinic, and the MIT CSAIL-Wistron Program.

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

# A  APPENDIX: ADDITIONAL RESULTS

## A.1  ADDITIONAL SYNTHETIC DATA VISUALIZATIONS

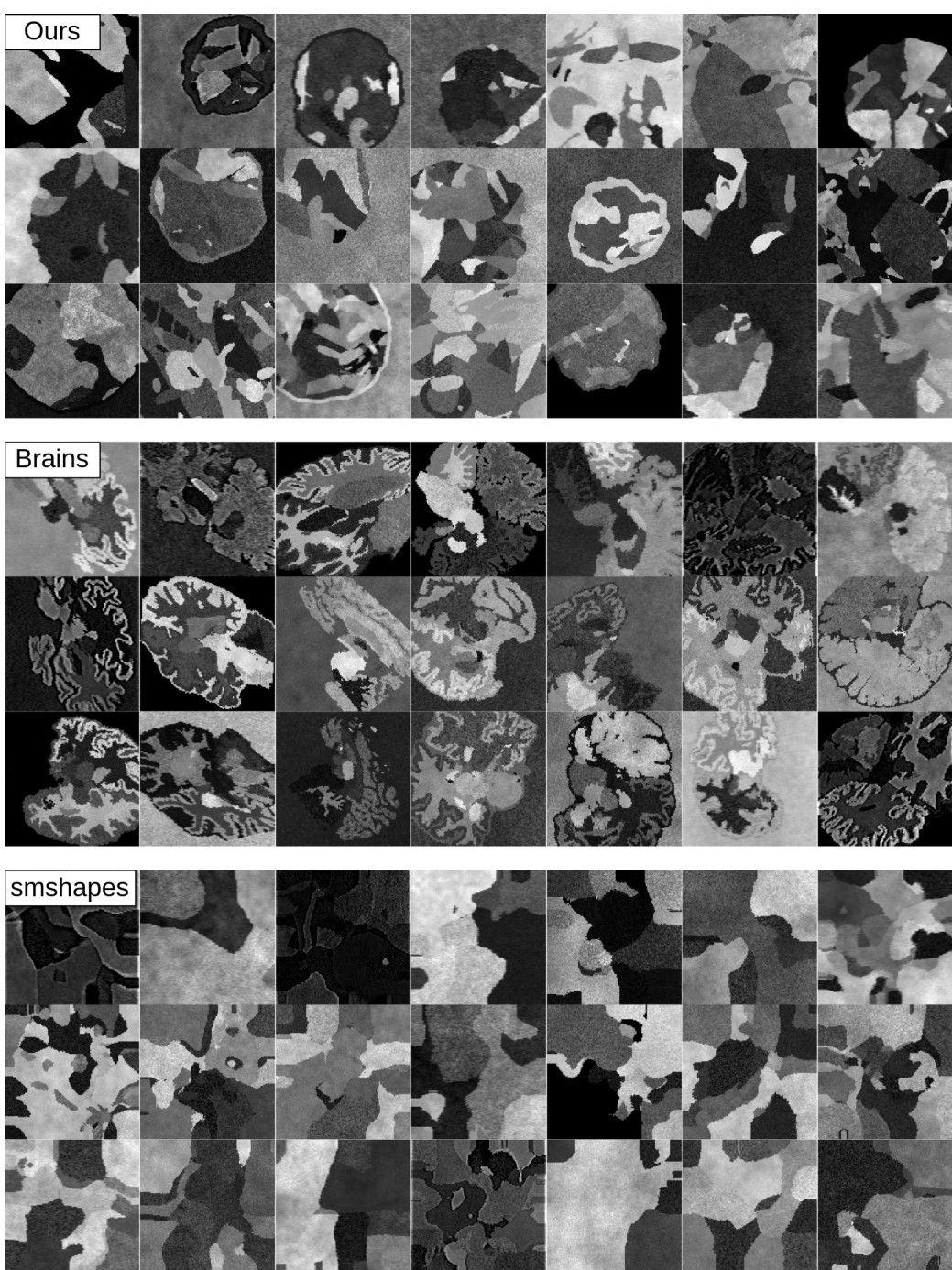

Figure 9: **Randomly selected synthetic volume (center slice) visualizations** sampled from our proposed data engine (**top**). For clarity, these samples have not had the online augmentations in Figure 12 applied to them. In the `Brains` and `smshapes` rows, we present samples corresponding to our ablations where we replace our proposed 3D label ensemble generator with real 3D brain labels (**middle**) or synthetically generated labels with no biomedical priors (**bottom**), respectively, but keep the appearance model unchanged.

## A.2 REGISTRATION REGULARIZATION WEIGHT GRID SEARCHES

Deformable registration optimization faces a trade-off between accuracy and warp field regularity, which is often tackled by using various regularizers and hyperparameters (Hoopes et al., 2021). For a fair comparison with `ConvexAdam`, we separately tune both the original framework and our extension (ConvexAdam-Ours) via a grid search over four hyperparameters on the validation splits of both datasets (i.e., L2R-Abdomen MRCT and MM-WHS). These hyperparameters include: Adam (Kingma & Ba, 2014) optimization grid spacing: $\{1, 2\}$, the warp smoothness penalty $\lambda$: $\{0.25, 0.5, 0.75, \dots, 2.5\}$, grid spacing: $\{2, 3, \dots, 6\}$ and `disp_hw`: $\{1, 2, \dots, 5\}$. All hyperparameters are tuned such that post-registration volume overlap (Dice) is maximized while maintaining deformation folds below $0.5\%$.

In Table 4, we summarize the Dice and folding statistics over the validation set grid searches. The reported means are averaged across subjects and hyperparameter configurations while standard deviations indicate inter-configuration spread. Using our network features (ConvexAdam-Ours) leads to substantially better performance and lower sensitivity to hyperparameter settings. This is corroborated in Fig. 10, where for different settings of $\lambda$ along the x-axis, we visualize Dice and folding voxel percentages for each hyperparameter configuration. We again find better performance at lower folding percentages while maintaining a lower sensitivity to hyperparameter settings.

Table 4: **Registration validation set grid search summary statistics (mean ± std.).** Here, the means are computed over all subjects and all hyperparameter configurations and the standard deviations correspond to the spread over all hyperparameter configurations.

| Method | L2R-Abdomen MRCT | | MM-WHS | |
| --- | --- | --- | --- | --- |
| | Dice ($\uparrow$) | Folds% ($\downarrow$) | Dice ($\uparrow$) | Folds% ($\downarrow$) |
| ConvexAdam-Ours | **0.863 ± 0.016** | **0.693 ± 1.423** | **0.661 ± 0.030** | **0.496 ± 1.249** |
| ConvexAdam | 0.806 ± 0.038 | 2.269 ± 3.292 | 0.652 ± 0.023 | 1.715 ± 3.572 |

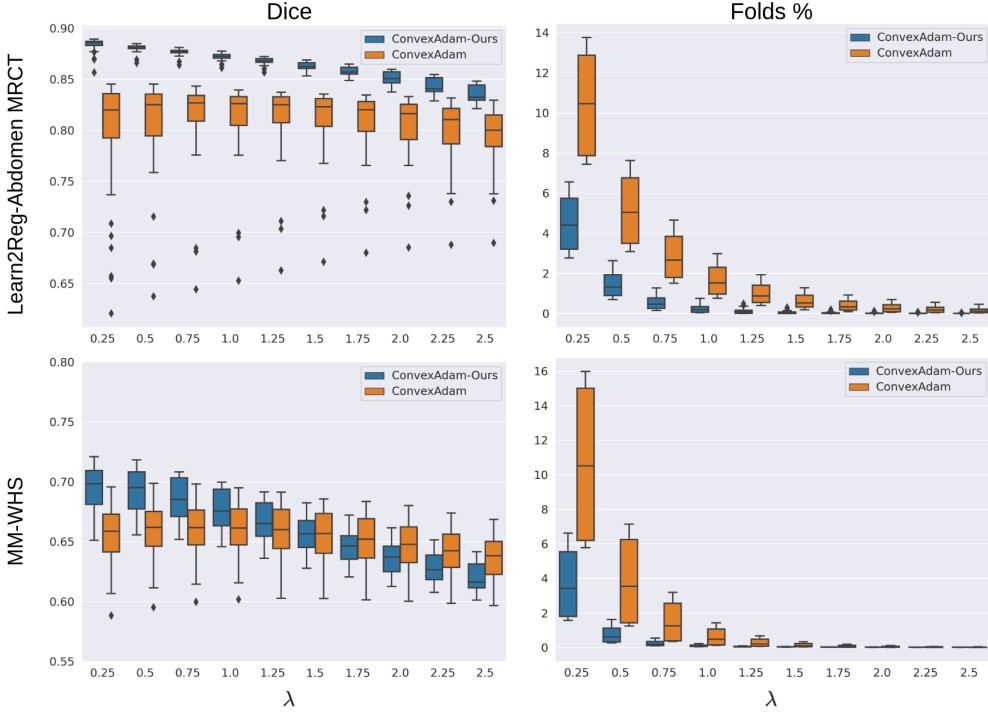

Figure 10: **Registration validation set grid search sweep statistics.** Here, each point contained in a boxplot is the average Dice for a particular hyperparameter configuration, given a fixed $\lambda$.

## A.3 ADDITIONAL REPRESENTATION VISUALIZATIONS

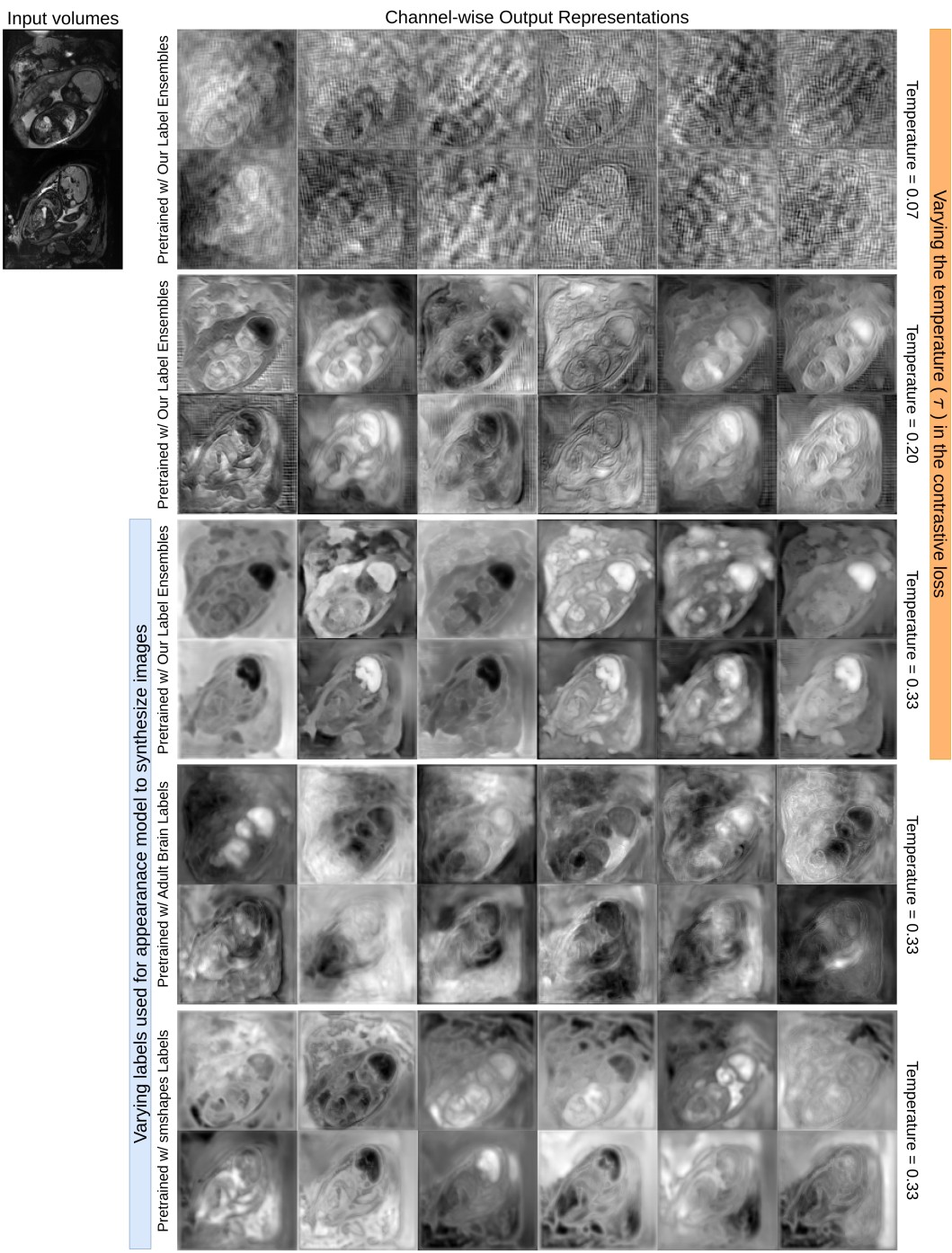

Figure 11: **Companion figure to Fig. 7: Feature visualizations for varying pretraining configurations**. Here, we visualize channel-wise output representations from our pretrained network for the two volumes at the **top left** for the first six channels. First, varying the temperature hyperparameter in the contrastive loss can lead to substantial aliasing (grouped rows 1–3). Then, changing our label ensemble synthesis model for other label sources (grouped rows 3–5) shows a loss in stability to nuisance variation. Our proposed model in grouped row 3 achieves both interpretable and stable representations on highly challenging volume pairs.

Table 5: Experiments comparing the multitask capabilities of the second-best segmentation model in Table 1 (PrimGeoSeg) **when matched for parameters and network architecture**, as measured by subject-averaged Dice coefficients and their corresponding bootstrapped std. deviations.

| Method | Architecture | Params. | MSD-Heart | Few-shot Segmentation | | | | Registration w/ ANTs | |
| | | | | L2RAb-MRI | FeTA | AMOS-CT | WUFetal | L2RAb | MMWHS |
|---|---|---|---|---|---|---|---|---|---|
| PrimGeoSeg | SwinUNETR | 67.2M | .87(.01) | .84(.05) | .79(.03) | .63(.01) | .76(.02) | .46(.09) | .51(.02) |
| PrimGeoSeg | UNet | 5.9M | .82(.02) | .84(.06) | .79(.03) | .56(.01) | .71(.02) | .36(.06) | .48(.02) |
| Ours | UNet | 5.9M | **.89(.01)** | **.86(.05)** | **.80(.03)** | **.61(.01)** | **.76(.02)** | **.70(.09)** | **.63(.02)** |

## A.4 NETWORK PARAMETER COUNT EFFECTS

Our experiments in Section 4.2 compare our pretrained U-Net with publicly released foundation models that have significantly higher parameter counts. To investigate the impact of network size, we use the same U-Net architecture as in our proposed model and retrain the second-highest average ranking method, PrimGeoSeg (Tadokoro et al., 2023), reducing its parameter count from 67.2M to 5.9M but matching all other training details to their code repository[1]. We then use it as a feature extractor for multi-modality registration with the `ANTs` solver (Tustison et al., 2020) and also finetune it for few-shot segmentation, as in our experiments in the main text.

As shown in Table 5, matching the parameters of PrimGeoSeg results in performance drops across 3 out of 5 few-shot segmentation datasets, as well as in both registration datasets. Notably, while PrimGeoSeg had originally outperformed our method on the AMOS-CT few-shot segmentation experiment in the main paper, its performance drops below ours by 5 mean Dice points when matched in parameter count, suggesting that the performance gap is a function of network size on that dataset.

## A.5 NEGATIVE RESULTS

While our proposed model consistently achieves state-of-the-art performance across several segmentation and registration benchmarks, it shows interesting trends on the preprocessed `neurite-oasis` (Hoopes et al., 2022a) T1w MRI neuroimage segmentation dataset, which is derived from the larger OASIS (LaMontagne et al., 2019) dataset. For this few-shot segmentation experiment, we train on $160^3$ crops, with a batch size of 3, and finetune on one annotated subject to segment the 35 classes provided by the dataset. In Table 6, we find that our proposed pretrained network does not improve over random initialization for this particular dataset.

Table 6: Few-shot 3D segmentation results on the `neurite-oasis` dataset (Hoopes et al., 2022a) reported as the mean Dice coefficient and bootstrapped std. deviation.

| RandInit | ModelsGenesis | MedicalNet | PrimGeoSeg | SMIT | DisruptiveAE | Ours |
|---|---|---|---|---|---|---|
| .82(.01) | **.84**(.01) | .75(.01) | .84(.01) | .83(.01) | .80(.01) | .82(.01) |

This result is consistent with the literature on the limited benefits of representation learning for few-shot adult neuroimage segmentation on OASIS in terms of Dice coefficient gains, as also reported in Ren et al. (2022). These trends may stem from the relatively high resolution, tissue contrast, and low inter-subject variability in OASIS. Further supporting this, prior work (Lee et al., 2019) has also shown that training adult neuroimage segmentation models from scratch on very small datasets can yield competitive results. Of the 8 remaining segmentation and registration tasks, our model either achieves the best (7 of 8 tasks) or second-best (1 of 8 tasks) performance, demonstrating its overall robustness.

---

[1]`https://github.com/SUPER-TADORY/PrimGeoSeg`

## A.6 Additional results on ultrasound and microscopy datasets

Table 7: Few-shot segmentation Dice results on 3D ultrasound (SegThy) and 3D microscopy (NucMM-M) datasets, reported as mean Dice score with bootstrapped standard deviations. The best-performing method is in **bold** and the runner-up is underlined.

| Method | SegThy | NucMM-M |
|---|---|---|
| Finetuning amount | 2 subjects | 1 volume |
| Number of classes | 3 | 1 |
| Random Init UNet | .82(.03) | .87(.03) |
| Transfer Learning | .81(.02) | .88(.02) |
| Models Genesis | .78(.02) | .83(.02) |
| MedicalNet | .77(.03) | .88(.03) |
| PrimGeoSeg | .78(.03) | .88(.03) |
| SMIT | .78(.03) | **.91**(.01) |
| Disruptive AE | .74(.02) | .88(.01) |
| Ours | **.84**(.02) | .89(.02) |

The imaging modalities used in the datasets utilized for few-shot segmentation in Table 1 consist of CT and various MRI sequences. We extend our evaluation here to include two additional datasets representing distinct volumetric biomedical imaging modalities: 3D ultrasound and 3D microscopy.

For ultrasound, we use the SegThy dataset (Krönke et al., 2022) which provides ultrasound scans from 28 healthy volunteers imaging the left and right side of the neck for each subject and labels for the thyroid, carotid artery, and jugular vein. We resample all volumes to a common spacing of $0.24mm^3$ and use two subjects (i.e., four left/right volumes) for few-shot finetuning. We use 4 subjects for validation and 6 subjects for held-out testing. For microscopy, we use the NucMM-M (Lin et al., 2021) dataset of nuclei in the mouse visual cortex at $0.480 \times 0.510 \times 0.510 \mu m^3$ resolution. As NucMM-M is an instance segmentation dataset, we repurpose it for semantic segmentation by binarizing the instance labels for consistency with our other experiments. NucMM-M provides 8 annotated volumes, of which we use 1 for finetuning, 3 for validation, and 4 for held-out testing.

In Table 7, for SegThy, the proposed model improves upon all baseline pretrained models by a clear margin, demonstrating our model's usefulness to the unseen domain of 3D ultrasound. For NucMM-M, we achieve 2nd place performance to SMIT, albeit with low inter-method variability. However, as in Tables 1, 2 and 7, our method consistently outperforms SMIT across all other seven segmentation tasks and both registration tasks, highlighting its generalizability.

## A.7 Additional comparisons with 2D DINOv2

Table 8: Few-shot segmentation results (mean Dice score with bootstrapped standard deviations) comparing our method to slicewise 2D finetuning of DINOv2 (Oquab et al., 2023) finetuned with LoRA (Hu et al., 2021) and a trainable FPN decoder head and a randomly initialized 3D UNet.

| | MSD-Heart | PROMISE12 | L2RAb-MRI | FeTA | AMOS-CT | WUFetal |
|---|---|---|---|---|---|---|
| 2D DINOv2 w/ LoRA, FPN | .78 (.01) | .72 (.02) | .72 (.07) | .61 (.04) | .25 (.01) | .43 (.02) |
| 3D UNet (Randomly Init.) | .85 (.01) | .80 (.02) | .85 (.06) | .78 (.03) | .56 (.01) | .73 (.02) |
| 3D Ours | **.89**(.01) | **.85**(.01) | **.86**(.06) | **.80**(.03) | **.61**(.01) | **.76**(.02) |

While our primary segmentation baselines are 3D foundation models specifically trained for radiology, recent work (Cekmeceli et al., 2024; Song et al., 2024) demonstrates that 2D natural vision foundation models, such as DINOv2 (Oquab et al., 2023), can be used for medical image analysis as well. In particular, concurrent work by Cekmeceli et al. (2024) shows that DINOv2 can be adapted for medical image segmentation in a fully supervised, slice-by-slice setting. To this end, we additionally benchmark against adapting DINOv2 for few-shot segmentation across all datasets in Table 1.

For DINOv2, we use the pretrained `ViT-L/14 distilled` (with registers) encoder. The encoder outputs an image embedding and we adapt it for 2D multi-label segmentation by attaching a trainable feature pyramid network (FPN) decoder head. We also train the encoder DINOv2 weights using low-rank adaptation (Hu et al., 2021) for efficiency. Since the encoder expects 3-channel inputs,

Table 9: Few-shot segmentation results (mean Dice score with bootstrapped standard deviations) comparing our method with finetuning an abdominal CT pretrained segmentation model (SuPreM).

| Methods | MSD-Heart | PROMISE12 | L2RAb-MRI | FeTA | WUFetal |
|---|---|---|---|---|---|
| SuPreM | .85 (.02) | .75 (.02) | .84 (.05) | .79 (.03) | .71 (.02) |
| Randomly Initialized UNet | .85 (.01) | .80 (.02) | .85 (.06) | .78 (.03) | .73 (.02) |
| Ours | **.89 (.01)** | **.85 (.01)** | **.86 (.06)** | **.80 (.03)** | **.76 (.02)** |

single-channel medical images are repeated across channels. All other training details (e.g., training losses, iterations, learning rates) are matched to our 3D segmentation setup described in Appendix B.5 and we pick the weights with the best validation set performance to compute the test set scores below.

Table 8 shows that while the 2D foundation model can be finetuned slice-by-slice for medical image segmentation, our 3D biomedically-focused approach consistently outperforms it for few-shot segmentation across all six datasets and distinct application domains considered. We also find that training a well-tuned 3D UNet from random initialization generally outperforms slicewise application of a 2D natural image foundation model finetuned on that medical dataset, highlighting the need for natively 3D generalist approaches such as ours. This finding aligns with other works benchmarking 2D vs. 3D self-supervised segmentation approaches as well, as in Ren et al. (2022).

### A.8 ADDITIONAL COMPARISONS WITH A ABDOMINAL CT FOUNDATION MODEL TRAINED ON LARGE-SCALE DATA

In parallel to our work, a large-scale annotated collection of 9,262 abdominal CT volumes was released (Li et al., 2024a) and is the basis for high-performance segmentation foundation models for abdominal CT (Li et al., 2024b). While these models perform well within their domain, it is unclear whether their pretrained weights transfer to new and unrelated biomedical domains and imaging modalities, such as fetal MRI. In contrast, our method, trained on highly variable synthetic data, does generalize consistently across diverse, unseen radiological domains and tasks.

To directly evaluate the transferability of abdominal CT foundation models to broader biomedical applications, we finetuned the SuPreM UNet (Li et al., 2024b), pretrained on AbdomenAtlas, on all non-abdominal CT datasets considered in the main paper. All splits, augmentations, and other training details are consistent with the main body of the paper.

Table 9 presents few-shot segmentation results for SuPreM, our model, and a randomly initialized UNet (with a matched architecture to ours), all finetuned on out-of-distribution datasets. SuPreM does not transfer well to new imaging modalities and biomedical applications. In contrast, our representations, trained only on highly variable synthetic data, generalize consistently.

We note that in future work, our framework can also be trained with supplemental real volumes and speculate that training on a mixture of synthetic and large-scale real datasets will have synergistic benefits, potentially yielding both high in-domain and out-of-domain performance.

## B  APPENDIX: ADDITIONAL IMPLEMENTATION DETAILS

### B.1  DATA ENGINE DETAILS

Our data engine in Fig. 2 A & B has several components. Here, we describe low-level implementation details. We note that we make extensive use of the MONAI (Cardoso et al., 2022), TorchIO (Pérez-García et al., 2021), and scikit-image (van der Walt et al., 2014) libraries for both label and volume synthesis.

### B.1.1  LABEL SYNTHESIS

The pseudocode in Algorithm 1 summarizes the synthesis process for a single label volume depicted in Fig.2A. This process assumes the availability of a set $Y$ of binary segmentation labels for different organs, which serve as templates. Specifically, these segmentations are taken from version 1 of the publicly available TotalSegmentator dataset (Wasserthal et al., 2023) (CC BY 4.0 license). To incorporate binary labels with multiple connected components, we merge individual rib labels into a single binary label for all ribs and also similarly pool the individual vertebral labels.

In Algorithm 1, $p_{\text{fg}}$ and $p_{\text{envelope}}$ refer to the probability of foreground masking and creating an envelope around the foreground, respectively, $w$ is the kernel width of the ball kernel used for morphological dilation and erosion, $\circ$ refers to a spatial deformation operator, and $*$ denotes the element-wise multiplication. The Perlin deformations $P_\sigma$ used are taken from Hoffmann et al. (2021).

---

**Algorithm 1** 3D synthetic label map $L$ generation

---

    **Input:** a dataset $Y$ of binary templates
    **Output:** Synthesized label map $L$

1: Initialize $L \in \mathbb{R}^{128 \times 128 \times 128}$ with all zero entries
2: Sample $N \sim \mathcal{U}\{20, 40\}$ templates $T = \{T_1, \ldots, T_N\}$ uniformly at random from $Y$

3: **for** $i = 1, 2, \ldots, N$ **do**
4:     Center-crop and pad $T_i$ to $(128, 128, 128)$
5:     Warp $T_i$ with random affine matrix $A_i$ s.t. $T_i \leftarrow T_i \circ A_i$
                $\triangleright$ translations and rotations are sampled from $\mathcal{U}[-5, 5]$ and $\mathcal{U}[-\pi, \pi]$, respectively
                $\triangleright$ scales and shears are sampled from $\mathcal{U}[-0.5, 0.5]$ and $\mathcal{U}[-0.5, 0.5]$, respectively
6:     Assign $L \leftarrow i * T_i$ at spatial indices where $T_i > 0$
7: **end for**

8: Median smooth $L$

9: **if** $p_{\text{fg}} > 1/3$ where $p_{\text{fg}} \sim \mathcal{U}[0, 1]$ **then**              $\triangleright$ Foreground mask
10:     Sample binary sphere $S \in \mathbb{B}^{128 \times 128 \times 128}$ with radius $r$ and center $c$
        where $r \sim \mathcal{U}\{48, 72\}$ and $c \sim \mathcal{U}\{32, 96\}$ independently along all axes
11:     Warp $S$ with Perlin deformation $P_\sigma$ s.t. $S \leftarrow S \circ P_\sigma$ where $\sigma \sim \mathcal{U}[1, 5]$
12:     Foreground mask $L$ as $L \leftarrow L * S$
13:     Increment $L \leftarrow L + 1$ at spatial indices where $S > 0$

14:     **if** $p_{\text{envelope}} > 0.5$ where $p_{\text{envelope}} \sim \mathcal{U}[0, 1]$ **then**    $\triangleright$ Create envelope around foreground
15:         Sample binary envelope $E \in \mathbb{B}^{128 \times 128 \times 128}$ where
        $E = \text{dilate}(S, w) \wedge (\neg(\text{erode}(S, w))$ where $w \sim \mathcal{U}\{2, 3, 4\}$
16:         Assign $L \leftarrow L + 1$ at spatial indices where $E > 0$
17:     **end if**
18: **end if**

---

### B.1.2  VOLUME SYNTHESIS

Given a label map $L$, we use it to conditionally sample two volumes/contrastive views $V_1$ and $V_2$ using an appearance model that is summarized in Fig. 12. Specifically, the two intensity volumes are generated by sampling from two independent Gaussian mixture models conditioned on the label map $L$. These preliminary 3D images are then independently transformed by a biomedical augmentation pipeline to form a contrastive pair of volumes. The term `Zero background` in Fig. 12 refers to setting intensities in the volumes spatially coinciding with the background label to 0.

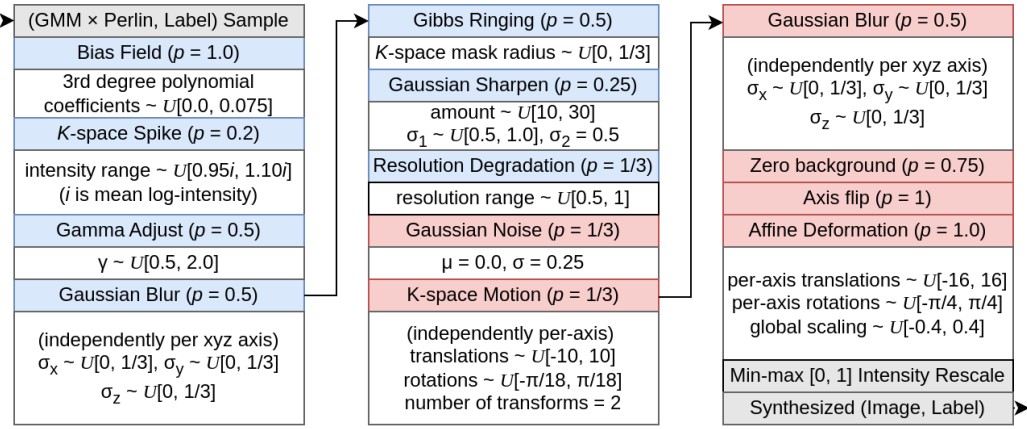

Figure 12: Post-Gaussian mixture model volume augmentation pipeline used to generate synthetic data for pretraining. Blue and red boxes refer to offline and online augmentations, respectively. All augmentations are applied to $128 \times 128 \times 128$ volumes. $p$ refers to the pre-defined probability of applying an individual augmentation and all other hyperparameters mentioned are consistent with MONAI (Cardoso et al., 2022) conventions. All augmentations use the default library hyperparameters besides Gamma Adjustments, Gaussian Blurs, and Gibbs Ringing, where we reduce the range of corruption to avoid degenerate samples.

As the combined pipeline of our label and volume synthesizer is computationally intensive, generating data on the fly could potentially bottleneck training. To mitigate this, we sample 120,000 3D label volumes and their corresponding 240,000 contrastive volume pairs offline using the proposed model. During training, we further apply additional online augmentations using a smaller pipeline. The offline and online augmentations are indicated in Fig. 12 by the blue and red boxes, respectively.

## B.2   NETWORK ARCHITECTURES

We employ a widely used (Billot et al., 2023b; Ren et al., 2022) UNet architecture for pretraining, fine-tuning, and for use as a feature extractor in our baseline comparisons and ablations. The architecture is described in Table 10. Each layer that contributes to the contrastive loss also includes an individual MLP network that projects sampled spatial indices onto an embedding space. Specifically, the architecture of each MLP consists of two `FC(128)-BN-ReLU` blocks (where `FC(w)` is a fully connected layer of width `w`), followed by an `FC(128)` layer and an $L_2$-normalization sequence at the final layer.

## B.3   PRETRAINING IMPLEMENTATION

We compute the contrastive loss in Eq. 1 on pre-activation convolutional features extracted from layers 7, 9, 12, 15, 18, and 23 in Table 10. Model selection is performed by tracking the validation contrastive loss to pick the best-performing checkpoint. We train using Adam (Kingma & Ba, 2014) with a starting learning rate of $2 \times 10^{-4}$ and a step decay towards 0 every 120,000 iterations.

## B.4   REGISTRATION EXPERIMENTS IMPLEMENTATION

Below, we first provide details regarding how the baselines were implemented in our registration experiments and then describe how our network features were integrated with existing solvers. For volumes of arbitrary grid sizes, we use sliding window inference with a window size of $128^3$ and an overlap ratio of 0.8. Further, we use region-of-interest masks for fixed and moving volumes whenever a registration method can use them.

Table 10: U-Net architectural details. We use the architecture from (Billot et al., 2023b; Ren et al., 2022). `Conv-BN-ReLU` refers to a sequence of 3D convolution with $3 \times 3 \times 3$ kernels, batch normalization, and pointwise ReLU activations. $n_c$ is the channel width multiplier and $n$ is the number of output channels. In our experiments, both $n_c$ and $n$ are set to 16.

| Layer index | Layer contents |
|---|---|
| 0 | Conv-BN-ReLU($n_c$) |
| 1 | Conv-BN-ReLU($n_c$) |
| 2 | Conv-BN-ReLU($n_c$) |
| 3 | MaxPool(2), Conv-BN-ReLU($2n_c$) |
| 4 | Conv-BN-ReLU($2n_c$) |
| 5 | MaxPool(2), Conv-BN-ReLU($4n_c$) |
| 6 | Conv-BN-ReLU($4n_c$) |
| 7 | MaxPool(2), Conv-BN-ReLU($8n_c$) |
| 8 | Conv-BN-ReLU($8n_c$) |
| 9 | MaxPool(2), Conv-BN-ReLU($16n_c$) |
| 10 | Conv-BN-ReLU($16n_c$) |
| 11 | Upsample $2\times$, Concatenate with layer 8 |
| 12 | Conv-BN-ReLU($16n_c$) |
| 13 | Conv-BN-ReLU($16n_c$) |
| 14 | Upsample $2\times$, Concatenate with layer 6 |
| 15 | Conv-BN-ReLU($4n_c$) |
| 16 | Conv-BN-ReLU($4n_c$) |
| 17 | Upsample $2\times$, Concatenate with layer 4 |
| 18 | Conv-BN-ReLU($2n_c$) |
| 19 | Conv-BN-ReLU($2n_c$) |
| 20 | Upsample $2\times$, Concatenate with layer 2 |
| 21 | Conv-BN-ReLU($n_c$) |
| 22 | Conv-BN-ReLU($n_c$) |
| 23 | Conv-BN-ReLU($n$) |

### B.4.1 BASELINE IMPLEMENTATION

**SynthMorph-shapes.** `SynthMorph-shapes` is a domain-randomized diffeomorphic registration UNet trained on synthetic volume pairs generated from label maps. It is optimized using a Dice loss subject to diffusion regularization and has no tunable hyperparameters at test time. We download its pretrained weights[2] from the VoxelMorph library and use their Tensorflow-based registration framework. Lastly, as their network is fully convolutional, we use the input volumes at their native resolution without resizing.

**uniGradICON/uniGradICON+IO.** `uniGradICON` is an approximately diffeomorphic registration foundation model trained on a variety of datasets. We use the binaries available on their repository[3] for our experiments. The off-the-shelf model does not have any hyperparameters at test-time and we use the default hyperparameters for the iterative variant (`uniGradICON+IO`).

**ConvexAdam.** `ConvexAdam` is a high-performance multimodality registration solver and we use the `b2671f8` commit of the repository[4]. Here we use the masked variant with default number of instance optimization iterations (80) and default hyperparameters of the MIND-SSC loss so as to use the same implementation of MIND-SSC across experiments. All the remaining parameters are fine-tuned on the validation data, see Appendix A.2 for more details.

**ANTs-MI.** The `ANTs-MI` baseline was run with the following command using the `ANTs` library:

```
antsRegistration \
```

---

[2] https://surfer.nmr.mgh.harvard.edu/ftp/data/voxelmorph/synthmorph/shapes-dice-vel-3-res-8-16-32-256f.h5

[3] https://github.com/uncbiag/uniGradICON

[4] https://github.com/multimodallearning/convexAdam/tree/b2671f86902390dec8dde702d0b583b451d84e98

```
--verbose 1 \
--dimensionality 3 \
--float 1 \
--output [OUTPUT_FOLDER/moved_, OUTPUT_FOLDER/moved_volume.nii.gz], \
--transform SyN[0.15] \
--metric MI[fixed_volume.nii.gz, moving_volume.nii.gz, 1, 48, Random, 0.666] \
--convergence 200x200x100 \
--shrink-factors 3x2x1 \
--smoothing-sigmas 3x2x0vox \
--interpolation Linear \
--masks [fixed_volume_mask.nii.gz, moving_volume_mask.nii.gz]
```

where `fixed_volume.nii.gz` and `moving_volume.nii.gz` are the input volumes to register and `fixed_volume_mask.nii.gz` and `moving_volume_mask.nii.gz` are binary masks indicating non-zero / non-background regions.

These settings correspond to using a three-level registration pyramid with the SyN algorithm and the mutual information loss for 200, 200, and 100 iterations at each level with corresponding level-specific smoothing kernels.

### B.4.2 MODIFICATIONS TO EXISTING REGISTRATION METHODS TO USE OUR NETWORK FEATURES

As our network produces 16 output channels, we modify existing registration solvers as below.

**ANTs.** To solve for a warp between a fixed and moving volume pair, we define 16 different MSE-based loss functions with ANTs. Specifically, each loss estimates the dissimilarity between corresponding channel volumes produced by our network for the fixed and moving inputs. We also downscale each individual loss by a tenth to trade off multiple data fidelity terms and regularization. The remaining modeling decisions and hyperparameters are identical to the baseline and use the following command:

```
antsRegistration \
--verbose 1 \
--dimensionality 3 \
--float 1 \
--output [OUTPUT_FOLDER/moved_, OUTPUT_FOLDER/moved_volume.nii.gz], \
--transform SyN[0.15] \
--convergence 200x200x100 \
--shrink-factors 3x2x1 \
--smoothing-sigmas 3x2x0vox \
--interpolation Linear \
--masks [fixed_volume_mask.nii.gz, moving_volume_mask.nii.gz]
--metric MeanSquares[fixed_ch1.nii.gz, moving_ch1.nii.gz, 0.1, 1, Random, 0.666] \
...
--metric MeanSquares[fixed_ch16.nii.gz, moving_ch16.nii.gz, 0.1, 1, Random, 0.666]
```

**ConvexAdam.** `ConvexAdam` already operates on multichannel inputs by using handcrafted MIND-SSC features. We therefore concatenate our network features with their original features and additionally multiply the network features by 0.1 for stable optimization. We perform a grid search over the same hyperparameters as the baseline `ConvexAdam` and set the remaining modeling decisions to be consistent with it.

### B.4.3 USING EXISTING 3D BIOMEDICAL SEGMENTATION FOUNDATION MODELS FOR REGISTRATION

In Section 4.3 and Table 2 of the main text, we demonstrate that current 3D biomedical segmentation models do not produce features that are directly usable by registration solvers such as `ANTs`. To elaborate, `MedicalNet` was excluded from analysis as it only provides a pretrained encoder without a decoder. `ModelsGenesis` produces one output feature map for an image that we use as is. The SwinUNETR-architecture based baselines (`PrimGeoSeg`, `SMIT`, and `DAE`) produce 48 output

channels that would necessitate prohibitively high compute costs at full-resolution. However, based on validation set performance, we find no significant difference between using ANTs with all 48 features or a subset of 16 (every fourth channel), and thus use the latter for all results reported for the test set. The ANTs hyperparameters for all experiments are the same as in App. B.4.2, varying only in the number of input channels corresponding to the output features produced by each method.

### B.4.4 MM-WHS DATA PREPARATION

**Prealignment.** The public proportion of the MM-WHS dataset consists of 20 unpaired and annotated 3D CTs and MRIs of the heart, all from different subjects. The CTs are high-resolution CT angiograms with tight fields of view around the heart, whereas the MRIs often include the subject's trunk. As our deformable registration baselines all assume affine pre-alignment, we align all the volumes to a common space. For accurate groupwise registration to this space, we formulate this as an affine atlas construction and registration problem (Avants et al., 2010).

All CT volumes are first clipped to [-450, 450] HU. We then arbitrarily select the first CT volume within the dataset (by subject ID) as an initial reference. We first resample it to a grid size of 160 × 160 × 128 at $1.142 \times 1.142 \times 1.283 mm^3$ resolution to define an initial coordinate system. This resampled volume is then used as an initial target for affine atlas construction with the remaining CT volumes. We use the following ANTs command[5] for groupwise affine alignment:

```
antsMultivariateTemplateConstruction2.sh \
  -a 2 \
  -d 3 \
  -A 0 \
  -o ${outputPath}T_ \
  -g 0.2 \
  -j 10 \
  -n 0 \
  -r 0 \
  -i 4 \
  -c 2 \
  -m MI \
  -l 1 \
  -t Affine \
  -q 100x50 \
  -f 4x2 \
  -s 2x1 \
  -b 1 \
  -y 0 \
  -z initial_target_ct.nii.gz \
  input_ct_*.nii.gz
```

Once an affine CT atlas is estimated, we then similarly register all MRI volumes to this CT atlas. In particular, due to the difficulty of intensity-based affine registration between the MRI and CT collections due to domain and FOV shifts, we estimate these affine transformations on the segmentations provided by the dataset and not the volumes themselves. Once all subject-to-atlas affine transformations are estimated on the segmentation volumes, they are used to warp all of the intensity volumes into the desired common space. All *deformable* registration experiments in our paper use only the intensity volumes and use the segmentations only for evaluation.

**Additional labels.** MM-WHS provides manual annotations for seven structures including heart chambers and portions of arteries for its original use in segmentation benchmarking. However, there are several anatomical structures in these volumes beyond the original labels such as the spine. Therefore, to better repurpose this data for holistic and non-local multi-modality registration evaluation, we annotate additional labels for the descending aorta and the spine. We use TotalSegmentator (Wasserthal et al., 2023) to segment these labels on the CT volumes and manually verify the results. For the MRI volumes, these new structures are annotated by a domain expert.

---

[5]https://github.com/ANTsX/ANTs/blob/master/Scripts/
antsMultivariateTemplateConstruction2.sh

Table 11: **Segmentation experimental dataset statistics.** All MRI modalities and sequences significantly differ from dataset to dataset.

|  | WUFetal | PROMISE12 | MSD-Heart | L2RAb-MRI | AMOS-CT | FeTA |
|---|---|---|---|---|---|---|
| Grid size | (112, 112, 80) | (320, 320, 24) | (320, 320, 115) | (192, 160, 192) | (512, 512, 115) | (256, 256, 256) |
| Original res. (mm$^3$) | (3.0, 3.0, 3.0) | (0.625, 0.625, 3.6) | (1.25, 1.25, 1.37) | (2.0, 2.0, 2.0) | (0.68, 0.68, 5.0) | (0.5, 0.5, 0.5) |
| Training res. (mm$^3$) | (3.0, 3.0, 3.0) | (0.625, 0.625, 0.625) | (1.25, 1.25, 1.37) | (2.0, 2.0, 2.0) | (1.5, 1.5, 2.0) | (0.5, 0.5, 0.5) |
| Training crop size | $80^3$ | $96^3$ | $96^3$ | $128^3$ | $96^3$ | $128^3$ |
| Training batch size | 4 | 4 | 4 | 4 | 4 | 4 |
| Num. of labels | 4 | 1 | 1 | 4 | 15 | 7 |
| Modality | BOLD MRI | T2w MRI | bSSFP MRI | SPIR MRI | CT | HASTE MRI |
| Finetuning vols. | 3 | 2 | 1 | 3 | 1 | 3 |
| Full supervision vols. | 60 | 50 | 6 | 24 | 180 | 40 |
| Validation vols. | 15 | 20 | 10 | 12 | 20 | 20 |
| Testing vols. | 24 | 30 | 4 | 12 | 100 | 20 |

## B.5 SEGMENTATION EXPERIMENTS IMPLEMENTATION

All image characteristics for each dataset are summarized in Table B.5. The datasets and modeling decisions were chosen to maximize diversity between the various segmentation settings.

For the publicly available datasets, we (re)split whatever annotated data is publicly available from the respective datasets. In particular, we resplit MSD-Heart, FeTA, and L2RAb-MRI's publicly available training sets to obtain training, validation, and held-out testing data. For AMOS-CT, we resplit the 200 volumes in the training set to obtain new training and validation splits and use their original and public 100 validation volumes as held-out testing data. For PROMISE12, we use the public splits, considering the `test` and `livechallengetest` splits to be its testing and validation splits, respectively.

PROMISE12 and AMOS-CT are highly anisotropic in resolution and are thus resampled to $0.625 \times 0.625 \times 0.625\ mm^3$ and $1.5 \times 1.5 \times 2.0\ mm^3$ resolution, respectively, with the latter resolution for AMOS-CT chosen to match the CT finetuning setting of the SwinUNETR baselines (Valanarasu et al., 2024). The CT intensities of AMOS-CT were clipped to [-450, 450] HU and the spatial grid extents of MSD-Heart and (post-resampling) PROMISE12 were padded to have a minimum of 96 slices.

Our UNet baselines use the crop sizes listed in Table B.5 and our SwinUNETR-based pretrained baselines use the crop sizes used for pretraining and finetuning in their respective original papers for both consistency and due to their pretrained transformer backbones. All augmentations are performed online and the MRI and CT dataset experiments in Table B.5 use the augmentations listed in Table 12 top and bottom, respectively. We found all baseline models and our model yielded better segmentation results when fully finetuned instead of other popular evaluation strategies for pretrained models such as linear probing. Additionally, we found that finetuning for a higher number of iterations with gradual learning rate decay consistently improved results on validation splits. Therefore, when finetuning, we train for 37,500 iterations with a batch size of four 3D crops using Adam with a starting step size of $2 \times 10^{-4}$ cosine decayed to 0. Finally, we exclude the background label when reporting Dice statistics.

## B.6 ALTERNATIVE LABEL SYNTHESIS MODELS

As in Sec. 4.3/label generation, we study the effect of our label ensemble synthesis model by replacing it with other label models used in biomedical image analysis. We detail our implementation of these alternatives below and visualize them in App. A.1.

**smshapes.** For `smshapes`, we follow the implementation of Hoffmann et al. (2021) for label generation with two key exceptions. First, for the label synthesis model, they fix the number of labels to synthesize to 26. For fair comparison with our framework which varies the number of labels in each volume, we modify smshapes to sample the same variable number of shapes. Second, for the appearance model, we match the hyperparameters of their GMM implementation to ours and also use

Table 12: **Augmentations** for segmentation fine-tuning experiments for MRI datasets (**top table**) and CT datasets (**bottom table**). Hyperparameters correspond to MONAI conventions (Cardoso et al., 2022).

| Prob. | MRI augmentation | Hyperparameters |
|---|---|---|
| 1.0 | Spatial crop | Crop size |
| 0.33 | Gaussian Noise | $\mu = 0.0, \sigma = 0.1$ |
| 0.33 | Bias field | coefficients $\sim \mathcal{U}[0, 0.075]$ |
| 0.33 | Gibbs ringing | $\alpha \sim \mathcal{U}[0, 0.33]$ |
| 0.33 | Gamma transform | $\gamma \sim \mathcal{U}[0, 4.5]$ |
| 0.33 | Gaussian blur | per-axis $\sigma \sim \mathcal{U}[0, 0.1]$ |
| 0.33 | Gaussian sharpen | $\alpha \sim \mathcal{U}[1, 30], \sigma_1 \sim \mathcal{U}[0, 3.0], \sigma_2 \sim \mathcal{U}[0, 1.0]$ |
| 1.0 | Affine warp | rotation$\sim \mathcal{U}[-\pi/4, \pi/4]$, scale$\sim \mathcal{U}[0.8, 1.2]$, shear$\sim \mathcal{U}[-0.2, 0.2]$ (all per axis) |

| Prob. | CT augmentation | Hyperparameters |
|---|---|---|
| 1.0 | Spatial foreground crop | Crop size, foreground label weight of 0.5 |
| 0.33 | Gaussian Noise | $\mu = 0.0, \sigma = 0.1$ |
| 0.33 | Gamma transform | $\gamma \sim \mathcal{U}[0, 4.5]$ |
| 0.33 | Gaussian blur | per-axis $\sigma \sim \mathcal{U}[0, 0.1]$ |
| 0.33 | Gaussian sharpen | $\alpha \sim \mathcal{U}[1, 30], \sigma_1 \sim \mathcal{U}[0, 3.0], \sigma_2 \sim \mathcal{U}[0, 1.0]$ |
| 1.0 | Affine warp | rotation$\sim \mathcal{U}[-\pi/4, \pi/4]$, scale$\sim \mathcal{U}[0.8, 1.2]$, shear$\sim \mathcal{U}[-0.2, 0.2]$ translation$\sim \mathcal{U}[-32, 32]$ (all per axis) |

multiplicative Perlin noise such that the appearance models are now matched and only the source of labels varies.

**Brains.** As opposed to synthesizing labels, the `Brains` experiment uses real brain labels in a manner similar to (Billot et al., 2023b). This is done to study the effect of synthesized label ensembles with randomized positions versus real biomedical anatomical configurations. We use 492, 500, and 581 T1-weighted brain scans from ADNI, HCP, and IXI, respectively, and segment them with SynthSeg (Billot et al., 2023b) to obtain training label maps. Using these labels, we sample 120,000 label volumes with 240,000 contrastive views as in our proposed model and match all other hyperparameters.

### B.7    Alternative pretraining losses

**Denoising pretraining.** For denoising pretraining, we maintain our data engine but pretrain the UNet to instead invert the intensity augmentations applied to the output of the initial Gaussian mixture model per sample, which is inspired by Iglesias et al. (2023). The UNet has a matched architecture to ours with an additional single convolutional layer mapping to a single-channel output volume. We train using the $L_1$ loss for denoising, match the optimization hyperparameters to our method, and select the checkpoint with the best validation $L_1$ loss.

**Removing label supervision.** As our data engine provides exact label supervision for each voxel, we use multi-positive label-supervised contrastive learning. However, several large-scale models are pretrained with self-supervised objectives not using any label supervision. To benchmark against these approaches, we use the self-supervised positive pair-only non-contrastive framework of Ren et al. (2021) with its losses applied to the same network layers as ours. Its variance and covariance loss weights are set to 0.01, the orthogonality weight is set to 100, and we halve the initial learning rate for stable training on our data as opposed to real brains used in their work.

### B.8    WUFetal dataset details

The in-house Whole Uterus Fetal (WUFetal) BOLD MRI dataset consists of 99 whole uterus volumes covering various pathologies, gestational ages, imaging artifacts, and the presence of twins. Due to this variability, this is a highly challenging dataset for few-shot segmentation. These scans were acquired on a 3T Skyra Siemens scanner using multi-slice gradient echo EPI sequences at 3mm

isotropic resolution (TR = [5-8] ms, TE = [32-38] ms, $\alpha = \pi/2$). All analyses were performed retrospectively on anonymized data and are IRB-approved.

