# OpenReview forum: "Learning General-purpose Biomedical Volume Representations using Randomized Synthesis"
_ICLR.cc/2025/Conference — ICLR 2025 Poster_

### Official Review · Reviewer_voNv · 2024-10-30

**Soundness:** 3
**Presentation:** 3
**Contribution:** 3
**Rating:** 8
**Confidence:** 4

**Summary:**

The paper proposes training a backbone that generalizes across different datasets using synthetically generated dataset. The proposed pre-training strategy has 3 main steps: 1) Given a large datasets of 104 annotated organs, randomly sample anatomies, deform them and create a volume by ensembling these anatomies, 2) add noise and other augmentations to the volumes that are sampled in the previous step to simulate realistic looking synthetic medical images from labels, 3) train a U-Net with a contrastive objective by sampling two volumes that share the same 3D semantic layout but differ in appearance, treating corresponding features at different encoder levels as positives and all others as negatives. The pre-trained backbone is validated on two different tasks: 3D registration and 3D few-shot segmentation; using multiple datasets.  The results show the effectiveness of the proposed backbone in the experiments compared to existing methods.

**Strengths:**

- Foundational models are showing promising performance lately; however, we lack of a 3D model that work across different modalities in medical imaging. The paper proposes a solution to this important problem using domain randomisation and contrastive learning.
- The paper contain experiments on multiple datasets both for registration and few shot segmentation, and the results demonstrate the potential of the method.
- The idea of combining the ideas of domain randomisation and local contrastive learning to train a generic 3D backbone is quite interesting and, to my knowledge, is novel.

**Weaknesses:**

- One issue I see in the paper is the convoluted description of the data engine step, especially the part creating the label ensemble model in section 3. I understand that this step is mainly based on the domain randomisation idea proposed in the literature. However, it is not really clear to me the steps between 201-207, especially the parts multiplying the masks with a randomly deformed sphere and randomly encasing half of the foreground-masked volumes.

- The images generated in the data engine step do not seem like as real medical images. Do they look like this because the deformation is too large? It is not clear why one would prefer training the model using such unrealistic images.

- The paper does not discuss the recent foundational models that show better generalization performance on many medical image datasets [1]. The downstream task performance of the representations obtained from the proposed backbone should be compared with the those obtained by the representations of a foundational model (e.g. DinoV2 [2]). For example, [3] is a recent paper that uses DinoV2 features for registration; but the same applies for the segmentation experiments. One can use the DinoV2 features for segmentation.

[1] Cekmeceli et al. "Do Vision Foundation Models Enhance Domain Generalization in Medical Image Segmentation?"
[2] Oquab et al. "DINOv2: Learning Robust Visual Features without Supervision"
[3] Song et al. "DINO-Reg: General Purpose Image Encoder for Training-Free Multi-modal Deformable Medical Image Registration"

**Questions:**

1- What are the exact steps of the "label ensemble model" described in Section 3? Please write elaborate description of these steps.
2- Why do the generated images not look like real medical images? Is it because the deformation is too large? Why such "unrealistic" looking images are preferred rather than more realistic ones obtained with smaller deformation?
3- How does the quality of the representations obtained by the proposed backbone compares with SoTA foundational models such as DinoV2 or SAM2?

---

> ### Author Response · Authors · 2024-11-21
> **Response to Reviewer voNv (part 1 of 2)**
>
> Thank you for the insightful comments and questions. We are happy to see that the reviewer found the work interesting and novel.
>
> > _”One issue I see in the paper is the convoluted description of the data engine step, especially the part creating the label ensemble model in section 3. I understand that this step is mainly based on the domain randomisation idea proposed in the literature. However, it is not really clear to me the steps between 201-207, especially the parts multiplying the masks with a randomly deformed sphere and randomly encasing half of the foreground-masked volumes.”_
>
> Thank you for raising this, we agree that it could be clearer and have edited the text accordingly. These steps are also described in detail in Appendix B.1.
>
> To summarize, many datasets in radiology consist of a foreground (body, head, etc.) and an empty background (e.g., air). Our data engine synthetically simulates these configurations by masking the synthetic volume with a random foreground mask (the deformed sphere). Further, many biomedical domains have a layer-like structure near the foreground (e.g., skin/fat in abdominal CT/MRI, the cortex in brain MRI, etc.). To simulate structures like these, we synthetically create a layer around the foreground as well.
>
> Quantitatively, we ablate this data engine choice in Table 3 (`Ours w/o FG mask`) and find a consistent drop in performance across both tasks and all four radiological datasets without it, highlighting the benefits of this step.
>
> > _”The images generated in the data engine step do not seem like as real medical images. Do they look like this because the deformation is too large? It is not clear why one would prefer training the model using such unrealistic images.”_
>
> We apologize for the potential confusion and clarify this point in the revision. The generated images are not intended to look like real medical images. Instead, they provide biomedically-informed generic training data that is densely labeled but not necessarily realistic. It is not because the deformation is too large.
>
> As intuition: Networks often overfit to both the intensity and shape characteristics of the training domain, limiting cross-domain generalization (e.g., CT to MRI segmentation).  While current domain randomized methods like [SynthSeg](https://www.sciencedirect.com/science/article/pii/S1361841523000506) show that training on synthetic brain images with random intensities yields contrast-invariant brain segmentation, these methods still overfit to the domain (i.e., brain label maps), need specific training frameworks for specific tasks (e.g., segmentation, registration), and need dense ground truth labels for any new application domain.
>
> Our work avoids overfitting to specific biomedical tasks and datasets by randomly sampling and deforming biomedical shape templates/primitives to construct label maps that are broadly generalizable across various biomedical contexts and tasks. Combined with our contrastive training mechanism that learns intensity invariance and a bias towards shape, we observe broad gains across datasets and tasks.
>
> Our ablations (Table 3) show that training on only real labels (`brains`) does not generalize to new biomedical domains. Further, training on purely synthetic labels (`smshapes`) does not work either, suggesting that biomedical shape priors are important for learning a robust representation. Our work balances between these extremes by instead independently deforming real biomedical shapes to achieve robust generalization across domains.

---

> > ### Author Response · Authors · 2024-11-21
> > **Response to Reviewer voNv (part 2 of 2)**
> >
> > > _”The paper does not discuss the recent foundational models that show better generalization performance on many medical image datasets [1]. The downstream task performance of the representations obtained from the proposed backbone should be compared with the those obtained by the representations of a foundational model (e.g. DinoV2 [2]). For example, [3] is a recent paper that uses DinoV2 features for registration; but the same applies for the segmentation experiments. One can use the DinoV2 features for segmentation.”_
> >
> > Thank you for the useful and relevant references! They are now cited and discussed in the revision.
> >
> > We clarify that [1] is concurrent work that differs from our approach and scope in many regards. [1] considers “domain generalization” to be finetuning 2D DINOv2 (+a decoder) on all available training data in one dataset and then assessing generalization to another related dataset in the same biomedical domain. For example, [1] trains on one prostate segmentation dataset and then tests generalization to another prostate dataset within the same imaging modality.
> >
> > While that is also an important problem, this is **unrelated to our scope**. Our form of domain generalization is about training a 3D network that has robust features for volumes in *any* new medical domain, whether it be prostates, fetuses, hearts, abdomens, etc., across any imaging modality. Further, [1] operates in the fully-supervised setting where they train on all available training data in a dataset. We instead operate in the context where only a few annotated volumes are available for finetuning.
> >
> > **Segmentation**: As suggested, we now finetune DINOv2 for multi-class semantic segmentation by finetuning the DINOv2-large encoder with registers (trained with [LoRA](https://arxiv.org/abs/2106.09685)) with an additional FPN decoder head. We train on 2D slices from our 3D datasets and present few-shot segmentation Dice results below.
> >
> > | Methods                       | MSD-Heart    | PROMISE12   | L2RAb-MRI   | FeTA        | AMOS-CT     | WUFetal     |
> > |-------------------------------|--------------|-------------|-------------|-------------|-------------|-------------|
> > | 2D DinoV2 + LORA + FPN       | 0.78(0.01)   | 0.72(0.02)  | 0.72(0.07)  | 0.61(0.04)  | 0.25(0.01)  | 0.43(0.02)  |
> > | 3D UNet (Randomly Initialized) | 0.85(0.01)   | 0.80(0.02)  | 0.85(0.06)  | 0.78(0.03)  | 0.56(0.01)  | 0.73(0.02)  |
> > | 3D Ours                       | **0.89**(0.01) | **0.85**(0.01) | **0.86**(0.06) | **0.80**(0.03) | **0.61**(0.01) | **0.76**(0.02) |
> >
> > We find that across all considered datasets, the proposed strategy widely outperforms finetuning DINOv2 for segmentation of 3D medical volumes in the low-data / few-shot regime. These experiments and further implementation details are now included in Appendix A.7.
> >
> > **Registration**: W.r.t. [3], it is a highly specific registration framework that uses DINOv2features. It finds that preprocessing axial 2D slices from rigid-aligned moving and fixed volumes by upsampling them to 5.3X their resolution, then running DINOv2 on every third slice (interpolating the rest), and finally performing a foreground-masked PCA of the DINOv2 features to get the first 24 principal components yields features suitable for their custom registration solver. We cannot directly compare against [3] as it does not share code nor crucial hyperparameters such as regularization weights, number of gradient iterations, etc.
> >
> > In contrast, our work requires no preprocessing, runs natively on 3D volumes, and is compatible with any already existing registration solver. We demonstrate that using our network features off-the-shelf with the commonly used ANTs and ConvexAdam solvers leads to 27 and 11 respective median Dice point improvements on Learn2Reg-MRCT and 5 and 7 respective points for MM-WHS.
> >
> > We have now cited and discussed [3] in the revision, thank you for bringing it to our attention.
> >
> > > _”Question: How does the quality of the representations obtained by the proposed backbone compares with SoTA foundational models such as DinoV2 or SAM2?”_
> >
> > For DINOv2, please see the discussion above. SAM2 is incompatible with the tasks and scope considered in our setting as SAM2 is not a general 3D feature extractor and requires user interactions/prompts to perform 2D interactive segmentation, whereas we perform *non-interactive* 3D registration and 3D semantic segmentation. Like DINOv2, it also returns image features at the 2D level and does not natively return volumetric features.

---

> > > ### Comment · Reviewer_voNv · 2024-11-22
> > > **Suggest acceptance for the paper**
> > >
> > > Thanks to the authors for the detailed response. I think this is a quite interesting work and has the potential to receive lots of attention from the medical imaging community. Therefore, I increase my score and suggest acceptance for this paper.

---

### Official Review · Reviewer_8WY8 · 2024-11-02

**Soundness:** 3
**Presentation:** 3
**Contribution:** 3
**Rating:** 6
**Confidence:** 4

**Summary:**

This work proposes a pre-training approach for downstream tasks related to fine-grained volumetric medical data: image registration and semantic segmentation. The authors propose to learn appearance invariance and shapes of human anatomy through synthetic dense pixel volumes. In this process, 3D volumes are synthesized by randomly recomposing 3D anatomical shapes and assigning multiple sets of random pixel values, in together with synthetic noises and deformations. Pairs of synthetic volumes are used for multi-scale contrastive learning. The proposed approach demonstrates improved image registration and low-shot image segmentation results compared to some previous works. Detailed ablation studies on the pre-training configurations toward downstream performances are conducted.

**Strengths:**

The overall methodology is straightforward and easy to understand. It echoes with the classical computer vision concept of invariance learning in the deep neural network era (although learned through a data-driven approach).

Improved image registration results on two public image registration benchmarks and image segmentation performance on six image segmentation datasets are shown, compared to those of some existing works.

The paper is well-written with sufficient clarity. The illustrations are self-explanatory. Readers will enjoy reading it.

**Weaknesses:**

Technical novelty: The core idea behind the approach is to leverage data-driven approach to learn invariance to pixel values through paired synthetic shapes and different pixel values, and to learn semantic-independent shape representation through random geometric (real-world and pure synthetic) shapes – both key ideas come from the well-established SynthMorph (random Ising shapes with random intensity + synthetic deformation for training registration networks. Hoffmann et al.) and SynthSeg (GMM-like pixel value model for repainting brain structures. Billot et al.) Despite leveraging more anatomical shapes beyond brain structures and applied to a contrastive framework, the essence remains unchanged.

Many medical decisions are made not only on shape but also on subtle textures, for example, differentiating subtypes of tumors/lesions – toward which the proposed over-simplified appearance model by nature falls short. More sophisticated texture models need to be carefully studied beyond this manuscript.

For the same reason, high-level global semantic information such as relative locations between anatomical structures cannot be learned due to the nature of this approach.

Real-world value: Given the increasing number of large-scale publicly accessible volumetric image datasets such as CT-RATE (Hamamci et al.), Totalsegmenter (Wasserthal et al.), and AbdomenAtlas (Li et al.), and the derived 3D foundation models, the real-world application of the proposed framework is unclear. Some of these large-scale public datasets come with fine-grain pixel-wise labels and associated radiological reports which provide additional supervision signals and text alignment potentials. The claimed generalization capability can be learned from multi-site large real-world datasets as well, through the intrinsic heterogeneity of big data and possibly through intense data augmentation.

**Questions:**

The proposed workflow involves many hyper-parameters (Figure 12) controlling the properties of generated synthetic volumes -- what is the rule of thumb for choosing them?

---

> ### Author Response · Authors · 2024-11-21
> **Response to Reviewer 8WY8 (part 1 of 3)**
>
> Thank you for your valuable and relevant questions and constructive comments. We are happy to see that the reviewer found the approach straightforward, motivated by classic computer vision, and well-presented, all leading to improved results across multiple datasets and tasks.
>
> We believe that there may be some miscommunications on our part and hope to address them below.
>
> > _”The core idea is … to learn invariance to pixel values through paired synthetic shapes and different pixel values, and to learn semantic-independent shape representation through random geometric shapes – both key ideas come from SynthMorph and SynthSeg … Despite leveraging more anatomical shapes beyond brain structures and applied to a contrastive framework, the essence remains unchanged.”_
>
> Thank you for raising this point. To clarify, SynthMorph and SynthSeg are highly specialized frameworks designed exclusively for brain registration and brain segmentation, respectively. Each network is trained in a fully supervised regime and is strongly biased towards the task and the anatomical region (the brain) it was trained to handle, as suggested by our experimental results (Table 3 rows 2–3 and Figure 5).
>
> Building networks that extract robust representations of biomedical datasets that are *task- and application-independent* requires a new approach, which we tackle with two contributions:
> - To ensure generalization to a wide range of shapes, we introduce a data engine that synthesizes random configurations of biomedical shape templates that yield robust *off-the-shelf* representations for any unseen anatomical region and imaging modality.
> - To provide generalization to a wide range of downstream tasks, we employ a contrastive loss that is supervised at every voxel and that utilizes paired samples from the data engine to learn invariance to common imaging variation and supports generalization across tasks.
>
> To our knowledge, both contributions are novel (as also noted by Reviewers `8tqR` and `voNv`) and essential for generalist performance.
>
> To empirically emphasize the distinctions from SynthMorph-shapes and SynthSeg:
> - Without our proposed data engine, contrastive training on only SynthSeg’s synthetic brain labels (Table 3, row 3, `brains`) or on only SynthMorph’s random shapes (Table 3, row 2, `smshapes`) drops performance consistently across all tasks and non-Brain datasets.
> - Further, in Figure 5, specifically for registration, our method outperforms the SynthMorph-shapes model trained on random shapes (mentioned by the reviewer) by large margins. On the abdominal dataset, there is a 72-point median Dice gap between SynthMorph-shapes and our method. On the cardiac dataset, there is an 11-point median Dice difference.
>
> Moreover, without our proposed contrastive training framework, training on entirely randomized shape configurations does not scale with other losses. In Table 3, “ablating pretraining losses”, we show that other pretraining losses such as denoising or unsupervised representation learning ([NeurIPS22](https://arxiv.org/abs/2206.04281)) result in worse performance.
>
> > _”Many medical decisions are made not only on shape but also on subtle textures, for example, differentiating subtypes of tumors/lesions – toward which the proposed over-simplified appearance model by nature falls short. More sophisticated texture models need to be carefully studied beyond this manuscript.”_
>
> To clarify, we do not aim to address tasks that require subtle texture-based differentiation such as lesion subtyping (this is listed in our limitations section).
>
> Importantly, this is an intentional choice as it has been established that networks that are biased towards shape instead of texture are more robust and more broadly applicable across domain shifts. For examples, see:
> - [ICLR’19 oral](https://arxiv.org/abs/1811.12231)
> - [NeurIPS’21 spotlight](https://openreview.net/forum?id=o2mbl-Hmfgd)
>
> Our work is similarly inspired. By focusing on learning biomedical shape representations, we achieve broad robustness to new biomedical domains and flexibility for multiple tasks, as illustrated by
> - SOTA multimodal registration across two datasets (abdominal MR/CT and cardiac MR/CT).
> - Consistent few-shot segmentation improvements across several completely distinct datasets featuring contexts such as fetuses, hearts, prostates, etc.
>
> As such, texture differentiation is out of scope for our paper which makes gains on several other fronts. We have now expanded the limitations regarding texture for clarity, thank you for bringing this up.

---

> > ### Author Response · Authors · 2024-11-21
> > **Response to Reviewer 8WY8 (part 2 of 3)**
> >
> > > _”High-level global semantic information such as relative locations between anatomical structures cannot be learned due to the nature of this approach.”_
> >
> > There are two points of miscommunication here which we hope to clarify.
> >
> > First, we **intentionally** remove global relative location information when sampling shapes from the TotalSegmentator (TS) dataset. Training on TS labels without our spatial reconfiguration will yield networks that fit the specific structure of the adult human torso/abdomen. However, these networks cannot then generalize when applied to new biomedical contexts (our main goal) such as whole uterus fetal imaging that bear no similarity to adult human torsos. We empirically demonstrate this phenomenon in Table 3, row 3 (`brains`) where we find that keeping the original relative locations between anatomical structures (as in the brain) does not yield useful representations for other non-brain domains.
> >
> > Second, all of **our registration experiments** show that relative locations between subjects can be estimated posthoc from the network’s representations. By definition, registration seeks to find correspondence between anatomical structures. As our network produces highly shape-biased and appearance-invariant representations, any existing registration solver can be used with the network’s representations to achieve state-of-the-art location-matching across modalities.
> >
> > > _”Given the increasing number of large-scale publicly accessible volumetric image datasets such as CT-RATE, Totalsegmenter, and AbdomenAtlas, and the derived 3D foundation models, the real-world application of the proposed framework is unclear.”_
> >
> > The datasets mentioned by the reviewer are abdominal/torso CT segmentation datasets. While they are the basis for excellent foundation models for abdominal CT (which are now cited), they do not necessarily generalize to new imaging modalities such as MRI (as noted by the AbdomenAtlas authors themselves in Appendix F.4 of [this paper](https://www.cs.jhu.edu/~alanlab/Pubs23/li2023suprem.pdf)) or to other biomedical applications such as fetal imaging.
> >
> > Quantitatively, our method already outperforms models that are trained on large public volumetric datasets when deployed to new domains and tasks. For instance, we outperform:
> > - [Disruptive Autoencoder](https://arxiv.org/abs/2307.16896), trained on 10,000+ MRI/CT volumes,
> > - and [SMIT](https://arxiv.org/abs/2205.10342), trained on 3600+ CT volumes,
> >
> > which are comparable to some of the datasets mentioned by the reviewer. Other baselines trained on mid-sized datasets (e.g., Med3D/MedicalNet and Models Genesis) similarly fall short of the performance we achieve with the proposed general representations.
> >
> > Importantly, **none** of these models can extract representations that are useful for other tasks beyond segmentation, as empirically verified in Table 2 for multimodal registration.
> >
> > To further address the reviewer’s suggestion w.r.t. large abdominal CT datasets, we now finetune the [SuPreM UNet](https://github.com/MrGiovanni/SuPreM) (pretrained on AbdomenAtlas and released by the AbdomenAtlas authors) on all non-abdominal CT datasets considered in the paper and present few-shot segmentation Dice results below.
> >
> > | Methods                       | MSD-Heart    | PROMISE12   | L2RAb-MRI   | FeTA        | WUFetal     |
> > |-------------------------------|--------------|-------------|-------------|-------------|-------------|
> > | SuPreM                        | 0.85(0.02)   | 0.75(0.02)  | 0.84(0.05)  | 0.79(0.03)  | 0.71(0.02)  |
> > | 3D UNet (Randomly Initialized) | 0.85(0.01)   | 0.80(0.02)  | 0.85(0.06)  | 0.78(0.03)  | 0.73(0.02)  |
> > | **Ours**                      | **0.89**(0.01) | **0.85**(0.01) | **0.86**(0.06) | **0.80**(0.03) | **0.76**(0.02) |
> >
> >
> > As expected, SuPreM (pretrained on AbdomenAtlas abdominal CT) does not generalize well to new *non-CT* imaging modalities and biomedical applications, whereas our representations, trained only on highly variable synthetic data generalize consistently.
> >
> > > _”The claimed generalization capability can be learned from multi-site large real-world datasets as well, through the intrinsic heterogeneity of big data and possibly through intense data augmentation.”_
> >
> > We respectfully disagree as no amount of intense data augmentation can make a large-scale multi-site abdominal CT dataset resemble a fetal MRI dataset. Further, to our knowledge, no work based on training on multi-site large real datasets has shown multi-task generalization capabilities for both 3D registration and segmentation in any biomedical domain.
> >
> > However, we see our synthetic approach as being fully compatible with these large-scale datasets as our framework can also be trained on a mixture of real and synthetic data. For a focused presentation, we only train on synthetic data and achieve consistent improvements over current work based on real datasets. However, in future work, we intend to mix both real and synthetic datasets.

---

> > > ### Author Response · Authors · 2024-11-21
> > > **Response to Reviewer 8WY8 (part 3 of 3)**
> > >
> > > > _”Some of these large-scale public datasets come with fine-grain pixel-wise labels and associated radiological reports which provide additional supervision signals and text alignment potentials. ”_
> > >
> > > To clarify, our work does not preclude also training on these datasets that come with associated text alongside images and segmentation labels. Our contrastive framework could simply use the proposed local contrastive losses on the real and synthetic images and use a CLIP-style contrastive text loss on only the real text and image pairs.
> > >
> > > However, using text supervision would constitute writing another paper and would change the focus of our current work. We focus on biomedically-informed synthetic data for domain generalization (across tasks and datasets) in medical vision. To this end, our work leads to the first 3D biomedical model that is capable of *both* achieving state-of-the-art registration and improving few-shot segmentation performance across several 3D datasets from any anatomical region, scale, or imaging modality (and not limited to only CT of the abdomen).
> > >
> > > > _”Question: The proposed workflow involves many hyper-parameters (Figure 12) controlling the properties of generated synthetic volumes -- what is the rule of thumb for choosing them?”_
> > >
> > > Figure 12 lists the image augmentations used in our work. They are standard operations such as Gaussian smoothing, Gamma adjustments, etc. used in all medical image analysis papers. Each augmentation has its own set of hyperparameters (e.g. sigma in Gaussian filtering) and we use the commonly used [MONAI](https://monai.io/) and [TorchIO](https://torchio.readthedocs.io/index.html) defaults for most. For a few augmentations (e.g., Gibbs ringing), we found the MONAI defaults to be too aggressive and to lead to degenerate outputs (e.g., all zero output with Gibbs augmentation when the sampled mask radius is close to 0), so we reduced the range when appropriate. We have added details on these choices in Appendix B.1.2.

---

> > > > ### Comment · Reviewer_8WY8 · 2024-11-24
> > > >
> > > > I have read through the responses and I would like to thank the authors for the clarifications.

---

> > > > > ### Author Response · Authors · 2024-11-28
> > > > >
> > > > > We have incorporated all of the reviewer’s remaining suggestions into the paper. Thank you for the response and the constructive feedback!
> > > > > > _”... claiming SynthMorph to be strongly biased toward brain images is a bit unfair as their … anatomy-free label generation is essentially similar to those employed in the manuscript.”_
> > > > >
> > > > > We apologize for our initial misunderstanding. In the revised PDF, we have accordingly expanded Section 2 “_Domain Randomization_” to discuss these points. To clarify, the anatomy-free label generation in SynthMorph is quite different from our approach. They sample multi-scale 4D Gaussian noise, deform it, and take a 1D argmax to create a synthetic 3D label map. As this approach does not incorporate any biomedical priors, our method (based on composing biomedical templates) outperforms it broadly in Figure 5 and Table 3. For additional context for readers, samples from these two approaches are also visualized in Appendix A.1/Figure 9.
> > > > >
> > > > > > _“The claim that shape-biased networks to be more robust than texture-biased counterparts may need to be handled with care. This would be the case for object recognition or anatomy segmentation but probably not the case for many medical decisions that actually rely on texture, such as lesion / tumor subtyping.”_
> > > > >
> > > > > We agree. This is highlighted as the first limitation in our limitations section. In the new revision, we have also added a reference to the need for texture analysis for tumor subtyping.
> > > > >
> > > > > > _”I do encourage the authors to supplement the above arguments (pros and cons for learning from real-world large-scale imaging datasets versus synthetic invariance/equivariance-steering datasets) into the manuscript.”_
> > > > >
> > > > > Thank you for the suggestion. Our submission discusses the pros and cons of large-scale real-world imaging datasets in the second paragraph of the Introduction (L037–046). Due to the hard 10-page limit, we have now incorporated the referenced arguments into Appendix A.8 of the revised PDF.
> > > > >
> > > > > Again, we thank the reviewer for their time and productive discussion.

---

> > > > > > ### Comment · Reviewer_8WY8 · 2024-12-02
> > > > > >
> > > > > > I have read through the follow up and I would like to thank the authors for the revisions. I have raised my score to the positive side.

---

> > > ### Comment · Reviewer_8WY8 · 2024-11-24
> > >
> > > I have read through the response and I would like to thank the authors for the discussion. I do encourage the authors to supplement the above arguments (pros and cons for learning from real-world large-scale imaging datasets versus synthetic invariance/equivariance-steering datasets) into the manuscript.

---

> > ### Comment · Reviewer_8WY8 · 2024-11-24
> >
> > I have read through the reponse and I would like to thank the authors for the discussion. With that being said, I would like to kindly remind that claiming SynthMorph to be strongly biased toward brain images is a bit unfair as there are two scenarios discussed in the paper: with and without anatomy labels. The anatomy-free label generation is essentially similar to those employed in the manuscript.
> > Also, the claim that shape-biased networks to be more robust than texture-biased counterparts may need to be handled with care. This would be the case for object recognition or anatomy segmentation but probably not the case for many medical decisions that actually rely on texture, such as lesion / tumor subtyping.
> > I would encourage the authors to better clarify their key distinctions beyond existing works such as SynthMorph and SynthSeg.

---

### Official Review · Reviewer_Emp1 · 2024-11-03

**Soundness:** 3
**Presentation:** 4
**Contribution:** 3
**Rating:** 8
**Confidence:** 4

**Summary:**

Authors present a method to generate highly variable synthetic imaging data which is then used to pre-train a 3D network using contrastive learning. The data generation method consists of drawing samples from a semantically labelled repository of biomedical shape templates to randomly populate an empty 3D volume. The volume is then deformed. Empty space and organ 'envelopes' are also simulated. To simulate different modalities and protocols, random intensity transformation is applied to the deformed 3D volume to yield 2 images. Typical imaging artifacts such a sbias field and blurring are simulated through random augmentations.The two images are fed into a UNet, and contrastive pre-training is performed on features at each decoder layer. An anchor point is chosen in one of the images, and all voxels of that label in both images are considered positive, and everything else negative pairs. The network yields features that can be used to finetune on other modalities and tasks. Importantly, the representations are modality-agnostic and anatomy-agnostic.

**Strengths:**

* The paper is very well written - it lays out the prior work and puts the contirbute in context.
* The approach yields representations that are both modality-agnostic and task-agnostic while removing the need for dataset-specific and anatomy-specific pre-training.
* Authors present results of several downstream tasks using their features including multi-modality registration image registration and few-shot segmentation on which their method outperform the others compared.
* Authors perform ablation studies on the various components of their pipeilne.
* The Authors present extensive visualization and quantitative results in their main text, and supplementary material. Algorithms and parameters are clearly presented too which allows for further scrutiny and improved reproducability.
* Authors are aware of the limitations of their approach and include these in the paper.

**Weaknesses:**

* The segmentation task performed using the Authors features may yield better results than the other methods that are compared, however the result still misses significant portions of the anatomical regions they aim to segment. The features require further adjustment and extensive fine-tuning to be useful in diagnosis and treatment.

**Questions:**

* Authors compare their randomized shape template-based synthetic data engine to one that uses data with no biomedical priors and one using brain regions. Can Authors elaborate more on the intuiton for why their randomly deformed template shapes are so effective? Is there some point at which the extent of the deformation causes the representations to be less useful?

---

> ### Author Response · Authors · 2024-11-21
>
> Thank you for the relevant feedback and questions and for emphasizing the work’s technical contributions and experimental thoroughness.
>
> > _”The segmentation task performed using the Authors features may yield better results than the other methods that are compared, however the result still misses significant portions of the anatomical regions they aim to segment. The features require further adjustment and extensive fine-tuning to be useful in diagnosis and treatment.”_
>
> We agree. However, this is true of all few-shot segmentation evaluation frameworks which, by definition, cannot match the performance of fully supervised networks finetuned on the entire training set. As we use only a few annotated volumes, we assess all pretrained initializations for data efficiency. We find that in the data-constrained setting, our work provides a robust dataset- and modality-agnostic improvement over previous work.
>
> This data efficiency in few-shot segmentation has several downstream benefits such as accelerating pseudo-labeling pipelines widely used to annotate vast datasets (e.g., [TotalSegmentator](https://arxiv.org/abs/2208.05868)), to provide a warm-start to active learning pipelines, among others. Further, although diminishing in extent, our method does yield benefits at higher levels of supervision as well, please see Figure 8 in the main paper.
>
> > _”Can Authors **elaborate more on the intuition** for why their randomly deformed template shapes are so effective?”_
>
> Thank you for raising this important question.
>
> Networks often overfit to both the intensity and shape characteristics of the training domain, limiting cross-domain generalization (e.g., CT to MRI segmentation).  While current methods like [SynthSeg](https://www.sciencedirect.com/science/article/pii/S1361841523000506) show that training on synthetic brain images with random intensities yields contrast-invariant brain segmentation, these methods still overfit to the domain (i.e., brain label maps), need specific training frameworks for specific tasks (e.g., segmentation, registration), and need dense ground truth labels for any new domain they are intended to be used in.
>
> Our work avoids overfitting to specific biomedical tasks and datasets by randomly sampling and deforming biomedical shape templates/primitives to construct label maps that are broadly generalizable across various biomedical contexts and tasks. Combined with our contrastive training mechanism that learns intensity invariance and a bias towards shape, we find broad gains across datasets and tasks.
>
> Our ablations (Table 3) show that training on only real labels (`brains`) does not generalize to new biomedical domains. Further, training on purely synthetic labels (`smshapes`) does not work either, suggesting that biomedical shape priors are important for learning a robust representation. Our work balances between these extremes by instead independently deforming real biomedical shapes to achieve robust generalization across domains.
>
> > _”Is there some point at which **the extent of the deformation** causes the representations to be less useful?”_
>
> The extent of deformation can hypothetically hamper downstream performance if the upper bound of the scale range of the affine deformations in the data engine is set too large. If it is too large for the last template sampled, the template can fill the entire volume, overwriting all others. In our work, the global scale range is set to [0.5, 1.5] and we did not observe this effect in practice.

---

> > ### Comment · Reviewer_Emp1 · 2024-11-22
> > **Acknowledgement of Author Response**
> >
> > Thank you to the Authors for their considered responses to this and the other Reviewers.
> >
> > Authors are mindful of the limitations of their work, and do not claim to be aiming for SoTA results in their few-shot learning regime compared with fully-supervised frameworks trained on large datasets. They are also clear in other responses that they do not aim to address every medical imaging task, e.g. those that require subtle texture-based differentiation. These nuances are perhaps where the concerns of Reviewer `8wy8` arise. While these are valid concerns, this Reviewer does not believe they detract from the novelty of the approach towards learning generalized representations of biomedical imaging data.
> >
> > This Reviewer maintains their score and encourages acceptance of the paper.

---

### Official Review · Reviewer_8tqR · 2024-11-04

**Soundness:** 3
**Presentation:** 3
**Contribution:** 3
**Rating:** 6
**Confidence:** 4

**Summary:**

The work proposes a new method for general (few-shot) medical image segmentation and registration. The method is based on generating realistic and varying synthetic data and training a generalist 3D segmentation and registration models on this data. The goal is to develop a method for generalizing to different imaging devices, medical procedures and conditions as well as different populations.

The data synthesis engine is capable to generate highly diverse samples and uses the totalsegmentattor dataset for shape templates. The “foundational” model is then pretrained using contrastive pretraining and finetuned in multi-modality registration and few-shot segmentation.

In experiments the authors demonstrate excellent performance in both downstream tasks.

**Strengths:**

-	Innovative approach with a highly powerful general synthetic data engine, I think this work adds a lot to the discussion on “foundational” models in medicine.

-	The method requires a comparatively small number of trainable hyperparameters compared to existing models while achieving a higher accuracy. This in my opinion is an important contribution towards sustainable models.

-	The authors provide the source code which is a major plus for reproducibility and discuss reproducibility aspects.

-	The authors evaluate the effect of different pretraining strategies in reasonable ablation studies.

-	The authors honestly show negative results in the appendix.

**Weaknesses:**

-	Considered datasets. If I am not missing any details the authors mostly evaluate their method on CT and MRI images. They even consider an MRI image as out of distribution. I think experimentation and ablation on more difficult and diverse datasets, such as 3D microscopy or Ultrasound, even if the results are not all positive would add to the discussion and validate the claim of a “general volumetric biomedical foundation model”

-	Framing of contribution. I would prefer if the authors tone down their wording about the model and clearly point out that it is a model for radiology or CT and MRI dataset and not a “general volumetric biomedical foundation model”.

-	Hyperparameter selection: What range of hyperparameters was tested, and how much time or resources were spent on tuning? How were the hyperparameters for the four baseline methods chosen? Especially for fine-tuning the baselines on your datasets, which I assume is done? Clearly describing the hyperparameter search is important for reproducibility. Please correct me if I was missing such details from the main manuscript.

**Questions:**

Please see weaknesses section.

---

> ### Author Response · Authors · 2024-11-21
>
> Thank you for the valuable feedback and for highlighting the method’s innovativeness, excellent downstream task performance, sustainability, reproducibility, and the paper’s directness with any limitations.
> > _“Considered datasets. If I am not missing any details the authors mostly evaluate their method on CT and MRI images.”_
>
> We apologize for the miscommunication on our part. To clarify, MRI is not a single modality and is a broad grouping of various imaging sequences that all highlight vastly different tissue properties.
>
> Our MRI datasets vary in sequences, using BOLD, HASTE, T2-weighted, bSSFP, and SPIR. For example, WUFetal uses BOLD MRI which measures blood oxygenation whereas SPIR is used for fat-suppressed structural imaging. Further, each dataset contains disparate anatomical regions and contexts, from pregnant uteruses to cardiac images. These differences make each dataset a unique domain for assessing generalizability.
>
> In our revision, we now explicitly clarify the sequence differences between datasets. Thank you for catching this.
> > _“They even consider an MRI image as out of distribution.”_
>
> If the reviewer is referring to L405, we meant that WUFetal (whole uterus BOLD images of fetuses, placentae, and surrounding organs) is not a publicly available dataset type, and thus it is out-of-distribution for our baselines pretrained on multiple public datasets. We have edited this sentence for clarity.
> > _’I think experimentation and ablation on more difficult and diverse datasets, such as 3D microscopy or Ultrasound, even if the results are not all positive would add to the discussion and validate the claim of a “general volumetric biomedical foundation model”’_
>
> We agree. As suggested, we have now added two new few-shot segmentation experiments on 3D Ultrasound and microscopy datasets into Appendix A.6, providing further details therein.
> - For 3D ultrasound, we use ultrasound volumes from the [SegThy](https://www.cs.cit.tum.de/camp/publications/segthy-dataset/) dataset which images the thyroid and provides labels for the thyroid, carotid artery, and jugular vein.
> - For 3D microscopy, we use the [NucMM-M](https://arxiv.org/abs/2107.05840) dataset that images cortical nuclei in the mouse visual cortex. Instance-level annotations were converted to binary semantic labels for consistency with the semantic segmentation approach used throughout the paper.
>
> For ease of reference, we summarize the Dice results below, with the best method in **bold** and runner-up in `code` formatting:
> |                    | SegThy | NucMM-M |
> |--------------------|--------|---------|
> | Finetuning amount | 2 subjects      | 1 volume      |
> | Random Init UNet  | `0.82`(0.03)   | 0.87(0.03)    |
> | Transfer Learning | 0.81(0.02)   | 0.88(0.02)    |
> | Models Genesis    | 0.78(0.02)   | 0.83(0.02)    |
> | MedicalNet        | 0.77(0.03)   | 0.88(0.03)    |
> | PrimGeoSeg        | 0.78(0.03)   | 0.88(0.03)    |
> | SMIT              | 0.78(0.03)   | **0.91**(0.01)    |
> | Disruptive AE     | 0.74(0.02)   | 0.88(0.01)    |
> | Ours              | **0.84**(0.02)   | `0.89`(0.02)    |
>
> For 3D ultrasound (SegThy), we improve upon all baseline pretrained models by a clear margin, demonstrating our model’s usefulness to this new domain. W.r.t. 3D microscopy (NucMM-M), we achieve 2nd place performance to SMIT. However, as in Tables 1 and 2 in our paper (plus SegThy), our method consistently outperforms SMIT across all other seven segmentation tasks and both registration tasks, highlighting its generalizability.
>
> To add to the discussion of claims, please see below.
> > _’Framing of contribution. I would prefer if the authors tone down their wording about the model and clearly point out that it is a model for radiology or CT and MRI dataset and not a “general volumetric biomedical foundation model”.’_
>
> We agree. We have now added clarifications that our model is specifically suited for radiology throughout the paper.
>
> > _’Hyperparameter selection: What range of hyperparameters was tested, and how much time or resources were spent on tuning? How were the hyperparameters for the four baseline methods chosen? Especially for fine-tuning the baselines on your datasets, which I assume is done? Clearly describing the hyperparameter search is important for reproducibility. Please correct me if I was missing such details from the main manuscript.’_
>
> These details are provided in the appendices, our apologies for the confusion.
>
> For the registration experiments, all baseline grid search design choices and results are in Appendix A.2. All registration baseline implementation details are in Appendix B.4.1–4.3. For the few-shot segmentation experiments, implementation, tuning, and hyperparameter details are provided in Appendix B.5. Lastly, our ablation implementation details are provided in Appendix B.6 and B.7. For further clarity, we have now added additional details where relevant.

---

> > ### Comment · Reviewer_8tqR · 2024-11-28
> > **Reviewer comment post rebuttal**
> >
> > Dear authors,
> >
> > I highly appreciate your detailed replies to my comments. I think the contributions are now clearly and fairly described in the manuscript. I have also read the other reviews and your comments. I find that others identified really sensible points, for example, in regard to SynthSeg.
> >
> > Overall, I am willing to keep my borderline rating, slightly leaning toward acceptance.

---

### Author Response · Authors · 2024-11-21
**Overall response**

We sincerely thank all reviewers for their insightful feedback and constructive suggestions. Their feedback has been incorporated to further improve the paper. All revised text in the manuscript appears in red.

We are happy that the paper was found to be technically innovative, interesting, and novel [`8tqR`, `voNv`]; well presented and enjoyable to read [`Emp1`, `8Wy8`]; and to make experimentally thorough progress on several downstream tasks and datasets [`8tqR`, `Emp1`, `8WY8`, `voNv`], all within a modality- and task-agnostic framework [`Emp1`, `voNv`].

To address specific concerns and incorporate reviewer feedback,
- Reviewer `8tqR`: We have conducted the suggested experiments with 3D ultrasound and microscopy datasets and have clarified that our contributions are radiology specific in the revision.
- Reviewer `Emp1`: We have elaborated further on the intuition behind our method and have provided further context regarding the gap relative to fully supervised segmentation.
- Reviewer `8WY8`: We have expanded on our technical distinctions concerning SynthSeg/Morph and have discussed limitations regarding application to texture-based classification. We have also clarified the new capabilities enabled by our approach over pretraining on large-scale real abdominal CT datasets and have included a new comparison to such approaches.
- Reviewer `voNv`: We have edited the data engine description and have addressed the realism of the synthetic samples. We have also conducted the suggested comparisons to recent 2D foundation models such as DINOv2.

All other points are addressed in the individual responses below. Again, we deeply appreciate the feedback and welcome any further discussion.

---

### Meta-Review · Area_Chair_2zJC · 2024-12-21

**Metareview:**

The submission presents a method for pre-training a generalist 3D backbone for medical image segmentation and registration using synthetic data, combining domain randomization and local contrastive learning. Reviewers praised the method's innovation, sustainability (fewer trainable parameters), and strong experimental validation across multiple datasets, with comprehensive ablations and reproducibility ensured by publicly available code. All reviewers agreed on the strong potential and significance of the work.

**Additional Comments On Reviewer Discussion:**

Concerns were raised about the limited evaluation on diverse modalities (e.g., ultrasound, 3D microscopy), unclear descriptions of the synthetic data generation process, and the unrealistic appearance of the generated data due to large deformations. Additionally, the method's focus on shape and appearance invariance was noted to fall short on capturing subtle textures or global semantic relationships crucial for some medical applications, and comparisons with SoTA models like DINOv2 and SAM2 were absent. Most concerns seem to be well addressed during the discussion.

---

### Decision · Program_Chairs · 2025-01-22

Accept (Poster)